# Explainable multi-task learning for multi-modality biological data analysis

Xin Tang [1,2,6], Jiawei Zhang [3,6], Yichun He [1,2,6], Xinhe Zhang [1], Zuwan Lin [4], Sebastian Partarrieu [1], Emma Bou Hanna[1], Zhaolin Ren[1], Hao Shen [1], Yuhong Yang[3], Xiao Wang [2,5], Na Li[1], Jie Ding [3] ✉ & Jia Liu [1] ✉

Current biotechnologies can simultaneously measure multiple high-dimensional modalities (e.g., RNA, DNA accessibility, and protein) from the same cells. A combination of different analytical tasks (e.g., multi-modal integration and cross-modal analysis) is required to comprehensively understand such data, inferring how gene regulation drives biological diversity and functions. However, current analytical methods are designed to perform a single task, only providing a partial picture of the multi-modal data. Here, we present UnitedNet, an explainable multi-task deep neural network capable of integrating different tasks to analyze single-cell multi-modality data. Applied to various multi-modality datasets (e.g., Patch-seq, multiome ATAC + gene expression, and spatial transcriptomics), UnitedNet demonstrates similar or better accuracy in multi-modal integration and cross-modal prediction compared with state-of-the-art methods. Moreover, by dissecting the trained UnitedNet with the explainable machine learning algorithm, we can directly quantify the relationship between gene expression and other modalities with cell-type specificity. UnitedNet is a comprehensive end-to-end framework that could be broadly applicable to single-cell multi-modality biology. This framework has the potential to facilitate the discovery of cell-type-specific regulation kinetics across transcriptomics and other modalities.

Recent advances in single-cell biotechnology make it possible to simultaneously measure gene expression along with other high-dimensional modalities for the same cells[1–3]. For example, the patch-seq technique simultaneously measures cell gene expression and intracellular electrical activity[4], and the multiome ATAC + gene expression technique jointly measures cell gene expression and DNA accessibility[5]. Such multimodal omics data provide direct and comprehensive views of cellular transcriptional and functional processes simultaneously. However, methods developed for analyzing single-modality biological data cannot be directly applied to multi-modality data[6]. Compared with single-modality analysis, recent studies have identified more tasks for multi-modality analysis[7] such as (i) identification of biologically meaningful groups from different modalities, enabling a deeper biological understanding of cellular identities and functions for different biological systems, and (ii) cross-modal prediction among different modalities, inferring the information of cells that cannot be easily or simultaneously measured. Moreover, the multi-modality data generated for the same type of cells provides the opportunity to discover cell-type-specific relationships between gene expression and other modalities, helping to uncover the regulatory mechanisms underlying the biological condition of interest. A method that can simultaneously address these different tasks and

[1]John A. Paulson School of Engineering and Applied Sciences, Harvard University, Boston, MA 02134, USA. [2]Broad Institute of MIT and Harvard, Cambridge, MA 02142, USA. [3]School of Statistics, University of Minnesota Twin Cities, Minneapolis, MN 55455, USA. [4]Department of Chemistry and Chemical Biology, Harvard University, Cambridge, MA 02138, USA. [5]Department of Chemistry, MIT, Cambridge, MA 02139, USA. [6]These authors contributed equally: Xin Tang, Jiawei Zhang, Yichun He. ✉e-mail: dingj@umn.edu; jia_liu@seas.harvard.edu

automatically quantify cross-modal relevance is needed to fully utilize the potential of multi-modality datasets.

Several methods of multi-modality analysis have been developed to address each task separately or to identify the cross-modal feature-to-feature relevance relationships. For the joint group identification task, multi-modality data integration methods have been developed to fuse different modality measurements into joint representations[8], which are then used for unsupervised or supervised classification to identify cell types and states or tissue regions[2,8–10]. For the cross-modal prediction task, autoencoder-based neural networks have been developed to predict across different modalities[11–16]. For the relevance discovery across modalities, Schema represents state-of-the-art multi-modal integration methods that can identify the features in a user-defined primary modality that are important to other modalities[17]. More recently, GrID-NET has also been proposed to identify genomic loci that mediate the regulation of specific genes in multiome ATAC + gene expression datasets[18].

Compared with the above methods, an approach that can address all tasks within a unified framework, quantify the cell-type-specific, cross-modal relevance, and do so without prior knowledge can streamline data analyses, potentially improve each task performance, and help gain biological insights from single-cell multi-modality data[19]. Still, combining multiple tasks into a single framework can be challenging for the following two reasons. First, each modality measurement has unique statistical characteristics (e.g., heterogeneous distributions and noise levels), requiring different statistical assumptions. While there have been several statistical models developed for different modalities (e.g., gene expression measurements[20–22]), a method that can accommodate unknown distributions of simultaneously measured modalities is still lacking. Second, joint group identification and cross-modal prediction typically represent separate objectives. Specifically, the objective of joint group identification is to penalize wrong group assignments of cells, whereas that of cross-modal prediction is to minimize the gap between predicted reconstruction and ground truth measurement. Thus, a strategy to integrate the different objectives needs to be designed to avoid performance degradation. Moreover, when no prior knowledge is given, it also remains a major challenge to find relevance relationships between gene expression and other modalities in certain cell types. If simply iterating all possible combinations of feature sets, the identification and quantification of the co-varying set of features will be computationally intractable for high-dimensional data. An efficient method is desired to first identify the set of features from multiple modalities that are important for a specific biological condition of interest (e.g., cell type) and then, quantify the relationship across these features.

Here, we introduce an explainable multi-task deep neural network to address the above challenges for multi-modality data analysis. This network has an encoder-decoder-discriminator structure and is trained by alternating between two tasks: joint group identification and cross-modal prediction. Specifically, this encoder-decoder-discriminator structure does not presume that the data distributions are known and instead implicitly approximates the statistical characteristics of each modality[23–25]. We have found that alternating training between joint group identification and cross-modal prediction maintains or improves the performance for both tasks. In addition, we applied explainable machine learning to dissect the trained network and quantify the cell-type-specific, cross-modal feature-to-feature relevance. We have applied this network to various multi-modality datasets (Fig. 1a), including (i) simulated multi-modality data with ground truth labels[26,27], (ii) simultaneously measured transcriptomics and intracellular electrophysiology[28,29] (multi-sensing data), (iii) simultaneously profiled transcriptomics and DNA accessibility[5,7] (multi-omics data), and (iv) spatially resolved transcriptomics and proteomics[30,31] (multimodal spatial-omics data).

The results show higher performances in both tasks, achieving similar or better unsupervised and supervised joint group identification and cross-modal prediction compared with other state-of-the-art methods. Moreover, we show that this approach recapitulates several previously published cell-type-specific feature-to-feature relevance relationships.

## Results

### UnitedNet: an explainable multi-task learning model for multi-modality biological data analysis

In this paper, we have developed UnitedNet, an explainable multi-task learning model to address the challenges discussed in the Introduction. Specifically, for joint group identification, UnitedNet uses encoders to obtain modality-specific codes (low-dimensional representations) and then fuses these codes into shared latent codes using an adaptive weighting scheme[32] (see "Methods"). The model then assigns the group labels, such as cell types, to each cell through unsupervised or supervised group identification networks, where the latter task is also known as annotation/label transfer[2,9,10] (Fig. 1b and Supplementary Fig. 1a). For cross-modal prediction, UnitedNet uses the encoder to obtain the source-modality-specific code and then predicts the data of the target modality through the target-modality decoder (Fig. 1b and Supplementary Fig. 1b). The discriminator networks are trained to distinguish between the data from the true modality and those reconstructed from the prediction, competing with the encoders and decoders in an adversarial manner to improve the accuracy of cross-modal prediction[23–25].

UnitedNet is trained using an overarching loss that consists of (i) an unsupervised clustering loss or a supervised classification loss that separates and closely packs shared latent codes in different clusters to better assign group labels[32,33], (ii) a contrastive loss that aligns the different modality-specific latent codes of the same cell and further separates the latent codes from other cells of different clusters[33,34], (iii) a reconstruction loss that compares the reconstruction from the encoders and decoders with the original data so that the latent code better represents the cell[15], (iv) a prediction loss that measures the performance of cross-modal predictions[15], (v) a discriminator loss that distinguishes the original and reconstructed data in the target modality[23–25], and (vi) a generator loss that pushes the decoded data to resemble the original data[23–25] (Fig. 1c). During the training, we optimize the network parameters by alternately training between the joint group identification and cross-modal prediction tasks, which are linked in the shared latent space (Fig. 1d, e).

In addition, as a trained UnitedNet combines information for both multimodal group identification and cross-modal prediction, dissecting it using post hoc explainable machine learning methods can reveal the cell-type-specific, cross-modal feature-to-feature relevance, which can help to facilitate the identification of biological insights from multimodal biological data. To do this, we apply the SHapley Additive exPlanations[35] algorithm (SHAP, see "Methods"), commonly used to interpret deep learning models, to dissect the trained UnitedNet. During the explainable learning, we can identify features that show higher relevance to specific groups (Fig. 1f) and then quantify the cross-modal feature-to-feature relevance within these groups (Fig. 1g).

### UnitedNet with multi-task learning exhibits robust and superior performance

To evaluate the performance of UnitedNet, we used a simulated dataset containing four modalities (DNA, pre-mRNA, mRNA, and protein) with their ground truth labels from Dyngen, a multi-omics biological process simulator[26] (Fig. 2a and see "Methods"). We first benchmarked the unsupervised joint group identification performance of UnitedNet against several state-of-the-art multi-modal integration methods, including Schema[17], Multi-Omic Factor Analysis

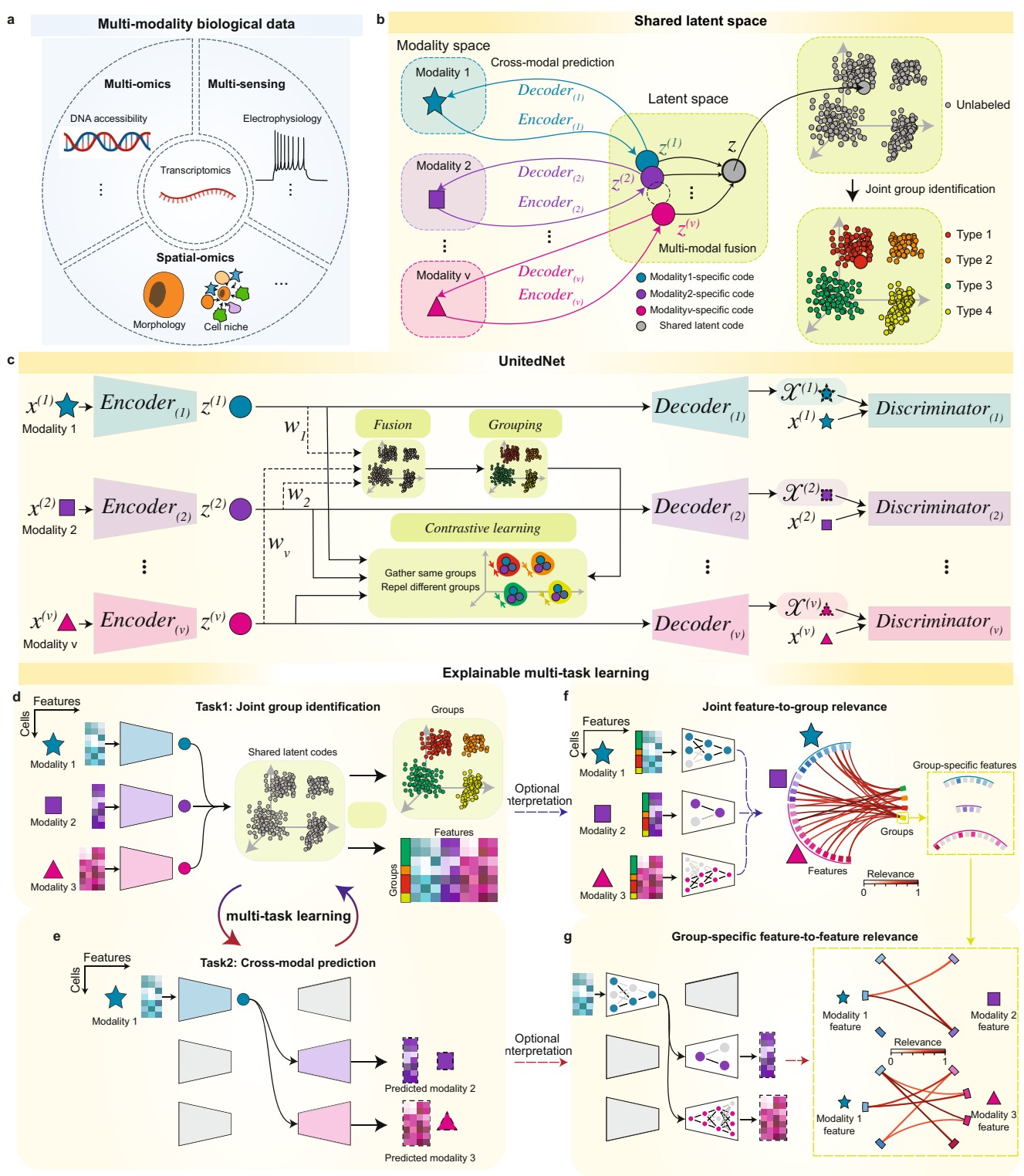

(MOFA)[36], totalVI[16], and Weighted Nearest Neighbor (WNN)[8]. We applied the Leiden clustering[37] method to cluster the integrated joint representations from these methods and used single-modality Leiden clustering as the performance baseline. As a result, UnitedNet consistently exhibits similar or better unsupervised joint group identification accuracy compared with the single-modality Leiden clustering and other state-of-the-art methods (Fig. 2b and "Methods"). We then performed an ablation analysis by removing the cross-modal prediction task in UnitedNet. Notably, we found that the unsupervised group identification accuracy decreased without multi-task learning (Fig. 2b

and "Methods"). Similarly, we evaluated the cross-modal prediction performance of UnitedNet through an ablation analysis. The results show that the ablation of either the multi-task learning or the discriminator (termed dual-autoencoder) reduced the average prediction accuracy of the network (Fig. 2c, "Methods"). Together, these benchmark studies and ablation analysis demonstrate the effectiveness of implementing the encoder-decoder-discriminator network structure and multi-task learning scheme for multi-modality data analysis.

We then studied why multi-task learning can improve the performance of both tasks. Based on the previous designation of

**Fig. 1 | Multi-task learning of multi-modality biological data by UnitedNet.**
**a** Schematics of representative multi-modality biological data: (i) simultaneously measured transcriptomics and intracellular electrophysiology (multi-sensing data), (ii) integratively profiled transcriptomics and DNA accessibility (multi-omics data), and (iii) spatially resolved transcriptomics and proteomics (spatial omics data).
**b** Schematics of designation of the shared-latent space. Different modality measurements from the same cell can be projected to a shared latent space as latent codes by encoders. The latent codes can be projected back to the modality space by decoders. In the latent space, latent codes representing different modalities from the same cell can be integrated as a unimodal shared latent code used for joint group identification. **c** Schematics showing the network structure of UnitedNet

based on the shared-latent space designed in (**b**). **d**, **e** Schematics showing the multi-task learning between joint group identification (**d**) and cross-modal prediction (**e**) by UnitedNet for analyzing multi-modality data. **f**, **g** Schematics showing the application of explainable learning methods to dissect the trained UnitedNet for identifying group-to-feature relevance (**f**) and the cross-modal feature-to-feature relevance (**g**). First, an explainable learning method dissects the encoders of a trained UnitedNet to identify the most relevant input features to each group (**f**). Then, within the grouped input features from (**f**), the explainable learning method further dissects the encoders and decoders of the trained UnitedNet to identify group-specific cross-modal feature-to-feature relevance (**g**). The drawings in panel **a** were created with BioRender.com.

shared-latent space in multi-modal and multi-task learning (Fig. 1b), we hypothesized that the joint training of joint group identification and cross-modal prediction tasks would reinforce each other through the shared latent space (Fig. 2a). To test this, we compared the shared latent codes learned from single-task training with multi-task training by UnitedNet using the simulated four-modality Dyngen dataset. The results show that compared with single-task learning (Fig. 2e, f), multi-task learning better aligned the modality-specific codes and better separated the clusters of the shared codes in the latent space (Fig. 2d). Together, they improved group identification efficiency and cross-modal prediction accuracy upon the model trained with single-task learning (Supplementary Fig. 2). We further quantified the relationship between the joint group identification and cross-modal prediction tasks throughout the training procedure. The results show that the performances of both tasks improved as the distance between the modality-specific codes decreased (Fig. 2g, h). Overall, the performances of group identification and cross-modal prediction tasks exhibit a positive correlation (Fig. 2i).

In addition, we demonstrated the robustness of UnitedNet when handling datasets with modality-specific noise. In applications, noises arising from different sources, such as the sequencing dropout effect and feature measurement error in multi-modality biological datasets, typically affect the network's performance. To address this challenge, UnitedNet applied an adaptive weighting scheme[32] to automatically assign lower weights to the modalities with more noise, reducing their impact on the group identification results (Supplementary Fig. 3a). To test the effectiveness of the adaptive weighting scheme in UnitedNet, we used simulated datasets with controllable dropout levels. Specifically, following a previous study[27], we simulated datasets with normal morphology modality data and noisy transcriptomics modality data at different controlled dropout levels[27] (Supplementary Fig. 3b) and applied UnitedNet to these datasets. The results showed that the modality weight of the transcriptomics modality increased as we decreased its dropout level, enabling a similar performance compared with the state-of-the-art methods (Supplementary Fig. 3c, d). Meanwhile, in the simulated four-modality Dyngen dataset without noises, the modality-specific weights were similar across four modalities (Supplementary Fig. 3e).

### UnitedNet provides accurate three-modality neuron type identification and cross-modal prediction for multi-sensing data
To demonstrate the ability of UnitedNet to analyze realistic multi-modality biological data, we applied it to the Patch-seq GABAergic neuron dataset, which measured morphology (M), electrophysiology (E), and transcriptomics (T) in the same neurons[29]. UnitedNet allowed for simultaneous unsupervised joint group identification and cross-modal prediction to identify cell types and predict modality-specific features, respectively (Fig. 3a).

We first benchmarked the unsupervised group identification performance of UnitedNet by combining cell electrophysiological and morphological features to identify transcriptomic cell

types[29,38]. Compared with other state-of-the-art cell typing methods including MOFA, totalVI, Schema, WNN, and Leiden clustering using single modality data, the performance of cell typing results by UnitedNet demonstrates improvement in terms of cell type separability and identification accuracy (Supplementary Fig. 4). We then benchmarked the cross-modal prediction performance between electrophysiology and transcriptomics modality by comparing UnitedNet with Coupled Autoencoder (CplAE), a deep neural network with an encoder-decoder structure[11,12]. UnitedNet achieved a similar or better performance in all directions of cross-modal prediction than CplAE with different hyperparameters (Supplementary Fig. 5).

Next, we performed simultaneous unsupervised joint group identification analysis and cross-modal prediction on the morphological-electrophysiological-transcriptomic (MET) datasets. By directly fusing the three modalities together and assigning the label for each cell, UnitedNet identified the cell MET-types with a high degree of congruence (ARI = 0.82) and a roughly diagonal correspondence (ARI = 0.41) between the major MET-types and subtle MET-types compared with the previously reported results (Fig. 3b–d). Furthermore, we benchmarked the performance of UnitedNet by comparing the results of MET-types clustering using Schema[17], MOFA[36], totalVI[16], and WNN[9] and Leiden clustering[37] using single modality data. The benchmarking results showed that UnitedNet performed similarly or better in terms of group identification accuracy compared to these methods (Supplementary Fig. 4). We also visualized learned modality-specific weights and found that the weights of gene expression were higher than other modalities in the Patch-seq dataset, which aligns with the previous understanding that gene expression is a more informative modality in the Patch-seq dataset[29].

For the cross-modal prediction task, previous methods such as coupled autoencoder were limited for the prediction between two modalities since they used an alignment loss function designed between two modalities, which cannot be directly applied to this three-modality dataset. In contrast, UnitedNet does not require explicit loss functions for modality alignment, thus allowing for the inclusion of multiple number of modalities as inputs. UnitedNet enabled the prediction of individual measurements across three modalities with high fidelity (Fig. 3e, Supplementary Fig. 6). We further examined the learned latent space of three modalities by the UnitedNet and found strong alignment between the transcriptomics and electrophysiology modalities (Supplementary Fig. 7a, b), which aligns with previous studies[11]. In addition, we found that the morphology modality was also aligned with the transcriptomics and electrophysiology modalities, with a relatively less level of alignment for the Pvalb neurons (Supplementary Fig. 7c). This relatively less level of alignment further supported the previous finding that, although Pvalb neurons have a similar gene expression profile, they exhibit both electrophysiological homogeneity and morphological diversity[39].

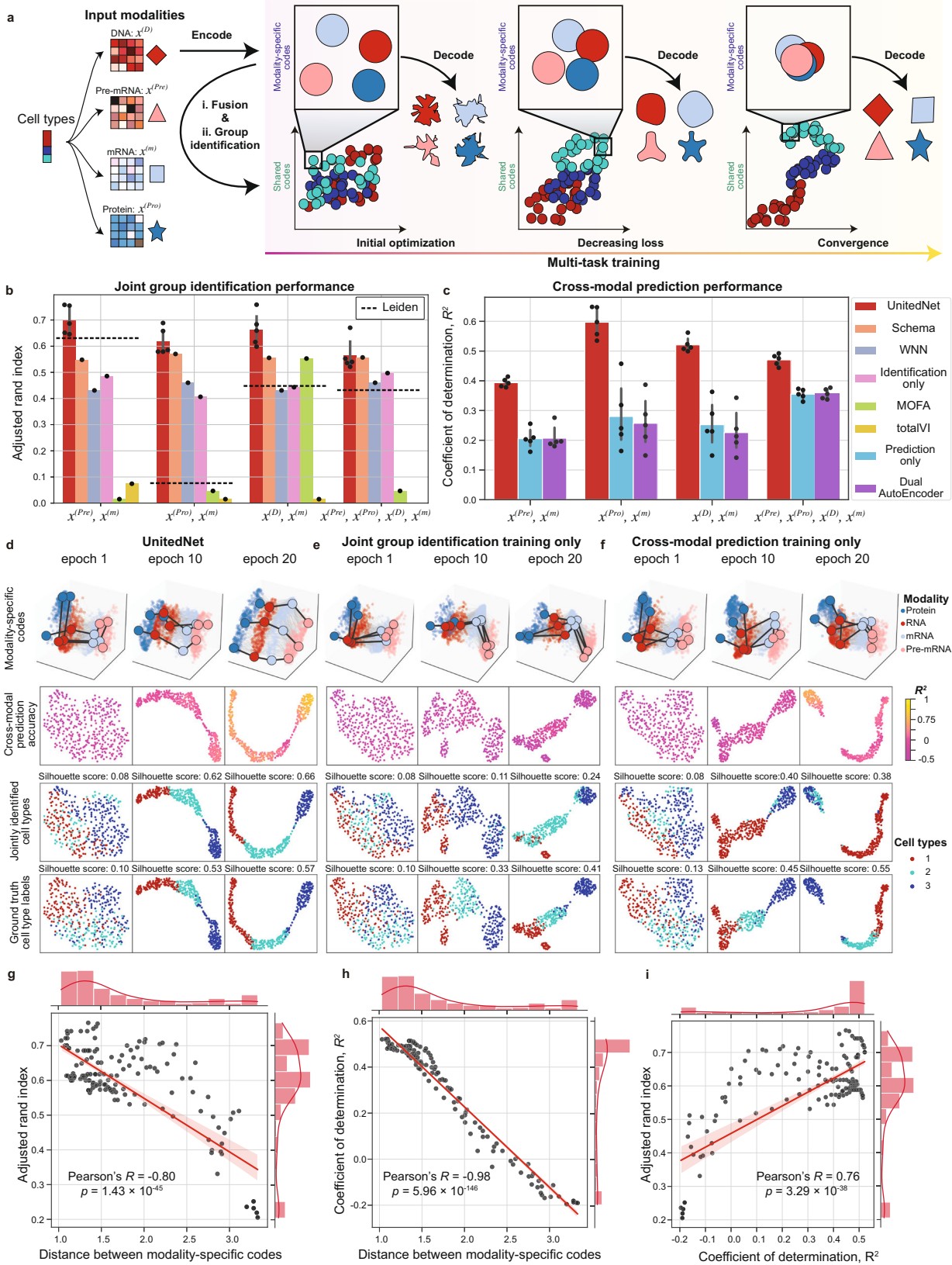

## UnitedNet indicates the neuron-type-specific, cross-modal feature-to-feature relevance relationship

We then dissected the trained UnitedNet using post hoc explainable learning, SHAP, to indicate the feature relevance in the Patch-seq GABAergic neuron datasets. Specifically, we used SHAP to assign the

importance value, known as the Shapley value, to each input feature with respect to any given model output such as a certain identified cell group or the cross-modality prediction of a certain feature[35]. By definition, features with high Shapley values are influential. Therefore, we chose features based on their ranking of the Shapley values.

**Fig. 2 | Performance evaluation of UnitedNet on a simulated Dyngen dataset.**
**a** Schematics of the optimization procedure of UnitedNet. $x^{(D)}$, $x^{(pre)}$, $x^{(m)}$, and $x^{(Pro)}$ represent the simulated modality of DNA, pre-mRNA, mRNA, and protein, respectively. Each modality measurement is encoded as a modality-specific code that is then fused as shared-latent codes. The performance of both the joint group identification and the cross-modal prediction task is enhanced as the modality-specific codes are aligned with each other. **b**, **c** Barplot reporting the performance comparison of joint group identification (**b**) and cross-modal prediction (**c**) of UnitedNet with those from other ablations. The dual autoencoder (dual AE) consists of two vanilla autoencoders without latent space alignment or discriminator. WNN, totalVI, Schema, and MOFA are conducted for the fusion of multiple modalities and then Leiden clustering is used for joint group identification on the fused representation. The dashed line in (**b**) represents the performance of group identification using Leiden clustering on the simulated modality of pre-mRNA, protein, DNA, and mRNA from Dyngen. $n = 5$ folds cross validation for panels (**b**) and (**c**). Data are represented as mean ± SD. **d**–**f** The latent space visualization by uniform manifold

approximation and projection (UMAP)[62] of UnitedNet (**d**) and other ablation versions in which we remove the training of cross-modal prediction task (**e**) or joint group identification task (**f**) with different training epochs for the Dyngen dataset with four modalities. The latent codes are colored to represent different prediction $R^2$ values (second row), cell type labels (third row), and ground truth labels (fourth row). Three representative codes in each of the modalities are highlighted in the first row to show the alignment between modalities. As the training epoch increases, the modality-specific codes are aligned with each other in UnitedNet (**d**) while in other ablations (**e**, **f**) the codes are misaligned. **g** The correlation plot between the inter-modality distance of modality-specific codes and joint group identification performance. $p = 1.43 \times 10^{-45}$. **h** The correlation plot between the inter-modality distance of modality-specific codes and cross-modal prediction performance. $p = 5.96 \times 10^{-146}$. **i** Correlation plot between cell typing performance and prediction performance. $p = 3.29 \times 10^{-38}$. Line of best fit shown in deep red and translucent bands around the regression line 95% confidence interval. Source data are provided as a Source data file.

To validate the robustness and effectiveness of this identification approach, we conducted the following experiments.

To evaluate the robustness, we considered (i) the inherent randomness of the SHAP method and (ii) the randomness introduced by training the UnitedNet model on different data, which are two major sources of randomness in the explainable learning outcomes. First, to validate the robustness of SHAP with different hyperparameters, we used the fixed Patch-seq GABAergic neuron dataset to train a UnitedNet model. Then, we applied 10 folds cross-validation to the Patch-seq GABAergic neuron dataset to calculate Shapley values. We then counted the frequency of the top n features identified by Shapley values in each cross-validation replication ($n = 7$ for each cell type in the Patch-seq GABAergic neuron dataset). The results (Supplementary Fig. 8) showed that the top 7 frequently selected features were consistently identified across the 10 cross-validation replications, which indicates the robustness of SHAP with respect to its inherent randomness.

Then, to further validate the robustness of SHAP in UnitedNet, we considered both the inherent randomness in SHAP and the randomness introduced by training UnitedNet on different data. Specifically, we used explainable learning to dissect UnitedNet models trained on different folds of cross-validation (Supplementary Fig. 9). Instead of using the entire Patch-seq GABAergic neuron dataset to train a single UnitedNet model, we divided the dataset into 10 folds, used every 9 folds to train a separate UnitedNet model and the remaining fold for testing. We then applied SHAP to dissect each model using the same methods described above. We counted the frequency of the top 7 features identified by Shapley values in each cross-validation replication. The results showed that explainable learning identified similar sets of features despite the inherent randomness of the SHAP method and the randomness introduced by training the UnitedNet model on different datasets (Supplementary Fig. 10). Taken together, these results support the robustness for dissecting the trained UnitedNet with SHAP.

Next, we quantitatively evaluated the effectiveness of the Shapley values and these SHAP-selected features (Supplementary Fig. 12). Given that neuron-type-specific features identified by previous studies are expected to be more biologically relevant, we hypothesized that the Shapley values of these features would be higher than those of randomly chosen features. Our results supported this hypothesis as we found higher Shapley values for marker genes compared with randomly chosen features in the Patch-seq GABAergic neuron dataset. Furthermore, we used the Shapley values as a predictor for marker genes. Our results showed that the marker features had higher predictability compared with randomly chosen features in the Patch-seq GABAergic neuron dataset (marker feature accuracy = 0.72 ± 0.07, mean ± SD; randomly chosen feature accuracy = 0.51 ± 0.03, mean ± SD, for 5 cell types * 3 modalities). These results demonstrate the

effectiveness of Shapley values in predicting group-specific features in multi-modality biology.

Then, using the Pvalb neuron type as an example, we qualitatively validated the SHAP-selected relevance (Fig. 4). For the group-to-feature relevance, SHAP successfully selected a subset of genes, electrophysiological features, and morphological features that are differentially expressed for Pvalb neurons (Fig. 4a, d–f). Investigating the Pvalb-neuron-specific cross-modal feature-to-feature relevance (Fig. 4b, c), we found that gene *Lrrc38* showed a higher relevance with the electrophysiological feature of the Pvalb neuron average firing rate during patch-clamp electrical stimulation using long square current steps. This result agreed with previous studies showing *Lrrc38*-related protein as one of the most crucial modulators for big potassium (BK) channels, which are critical to neuronal firing dynamics and neurotransmitter release[40,41]. These results suggested that UnitedNet could be potentially used to facilitate the identification of cell-type-specific gene-to-function relevance for Patch-seq data.

The above experiments showed the robustness and effectiveness of the Shapley value-based feature relevance identification. Based on the ranking and robustness of the Shapley value for each feature (Supplementary Figs. 9–10, see "Methods"), the final selected subset of neuron-type-specific genes, electrophysiological features, morphological features, and quantified neuron-type-specific, cross-modal feature-to-feature relevance are shown in Fig. 4 and Supplementary Fig. 11.

## UnitedNet enables accurate joint annotation transfer and cross-modal prediction in multi-omics data

We applied UnitedNet to analyze large-scale datasets from multiple batches of samples. These large-scale multi-omics datasets are typically generated to annotate cell molecular types from different batches of samples. For example, the single-cell transposase-accessible DNA sequencing technique[5] that combines gene expression and genome-wide DNA accessibility (namely, multiome ATAC + gene expression data) has been used to profile diverse types of immune cells[7]. One challenge in analyzing these datasets is using previously annotated multi-modality datasets as references to analyze new measurements from a different batch of the same biological system. Another challenge is the batch effect, which refers to differences between cells caused by inter-sample variations. This batch effect makes it difficult to use labeled multiome ATAC + gene expression datasets as a reference atlas to annotate other new datasets. We found that UnitedNet can address these challenges in two ways. First, it can perform a supervised group identification task (termed annotation transfer) that automatically identifies cell types in new, unlabeled test samples based on previously labeled training samples, while simultaneously enabling cross-modal prediction. In addition, we found that UnitedNet can

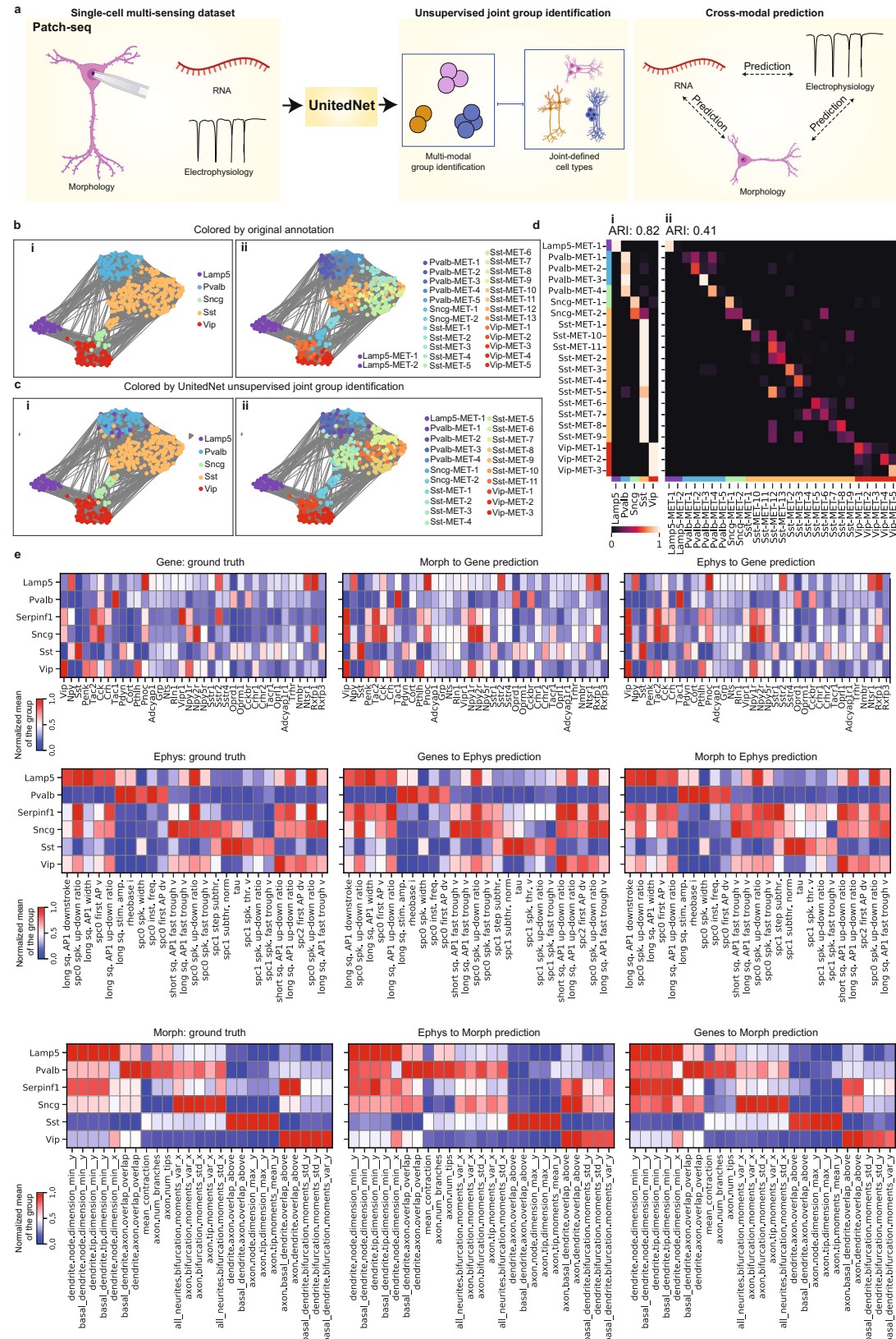

effectively reduce batch effects among different biological samples (Fig. 5a).

We applied UnitedNet to multiome ATAC + gene expression datasets measured from the bone marrow mononuclear cells (BMMCs) in 13 batches from different tissue sites and donors[7]. These datasets contain 22 previously identified and annotated cell types (Fig. 5b i–ii

and Fig. 5c i–ii). We trained UnitedNet using data from 12 batches of data with the corresponding cell type annotations serving as the labels from the training samples. We then tested the performance on the 13th batch. UnitedNet was able to successfully (i) integrate the 12-batch reference datasets and learn the 22 annotated cell types, (ii) simultaneously map the unlabeled test dataset (the 13th batch) into the same

**Fig. 3 | Multi-modal analyses of a multi-sensing dataset by UnitedNet.**
**a** Schematics illustrating the unsupervised joint group identification and cross-modal prediction of a Patch-seq dataset by UnitedNet. The patch-seq experiment simultaneously measures transcriptomics, electrophysiology, and morphology from the same cells to jointly define cell types. **b**, **c** UMAP representations of the shared codes in the shared latent space that are color-coded by the joint morphology-electrophysiology-transcriptomics (MET)-types labeled by the reference (**b**) and identified by UnitedNet (**c**), where MET major cell type annotations are in (i) and MET cell subtype annotations are in (ii). **d** Confusion matrices comparing (i) joint major cell types and (ii) cell subtypes between the reference labels and UnitedNet-identified labels. **e** Heatmap comparing cross-modal predicted gene expression, electrophysiological, and morphological features averaged over annotated major transcriptomics cell types with the ground truth. Source data are provided as a Source data file. The drawings in panel **a** were created with BioRender.com.

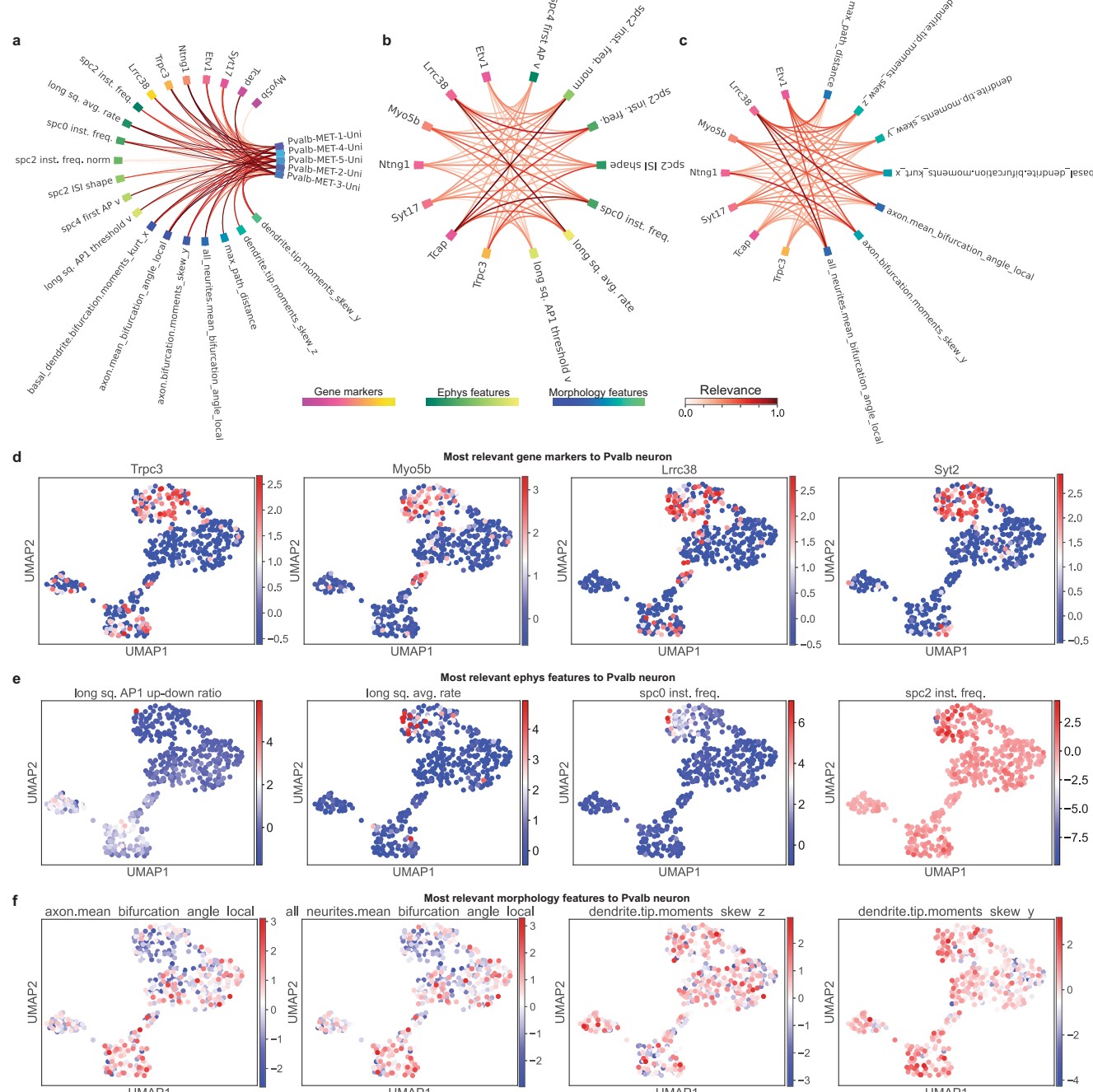

**Fig. 4 | Explainable learning for UnitedNet trained with patch-seq GABAergic neuron data.** **a** Chord diagram showing the feature-to-group relevance for Pvalb neurons. The relevance values are min-max normalized to 0-1. The 'Uni' represents the label predicted from UnitedNet. **b**, **c** Chord diagrams showing the gene-to-electrophysiology relevance (**b**) and gene-to-morphology relevance (**c**). The features used for cross-modal relevance analysis are from (**a**). The relevance values are min-max normalized to 0-1. **d**–**f** UMAP representations of the shared codes of patch-seq GABAergic neuron data in the shared latent space that are color-coded by most Pvalb-neuron-relevant features from the modality of gene expression (**d**), electrophysiology (**e**), and morphology (**f**). Source data are provided as a Source data file.

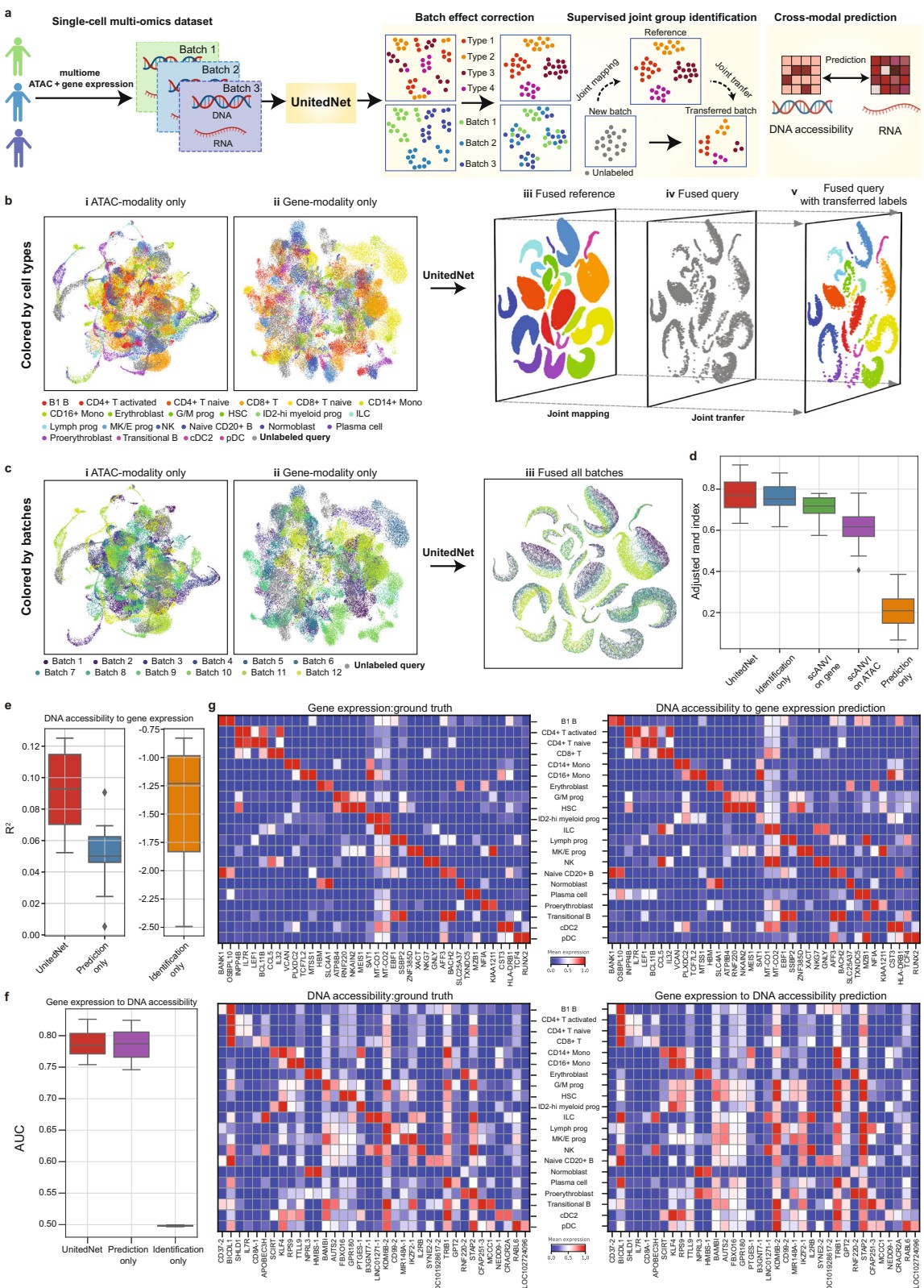

22 separated clusters, and (iii) reduce the batch effect (Fig. 5b iii–v and Fig. 5c iii). We also visualized the learned modality-specific weights and found that the weights of DNA accessibility and gene expression were similar, indicating the similar importance of these two modalities with respect to cell type identification (Supplementary Fig. 3f).

To understand how UnitedNet reduces batch effect, we conducted additional ablation studies (Supplementary Fig. 13). These studies showed that while the joint group identification task helped reduce the batch effect through its classification loss function, it also led to over-fitting of group identification. In contrast, the cross-modal prediction task improved group identification, but it could also contribute to a stronger batch effect. By combining both tasks in an alternating training approach, UnitedNet was able to leverage their strengths to reduce the batch effect while maintaining group identification performance.

**Fig. 5 | Automated multi-modal annotation transfer and cross-modal prediction of multi-omics datasets by UnitedNet. a** Schematics of a multiome ATAC + gene expression data analysis pipeline by UnitedNet. UnitedNet uses RNA and DNA accessibility data along with their cell type annotation labels as inputs to train the network. **b** Supervised joint group identification enabled by UnitedNet. The UMAP latent space visualizations show DNA accessibility (i), RNA (ii), and shared latent codes of UnitedNet (iii-v) of the 13-batch multiome ATAC + gene expression dataset. In this process, UnitedNet first fused the shared latent codes from two modalities and grouped the codes based on cell type annotations (iii). Then, it projected the unlabeled query batch to the learned shared latent space (iv), transferring the label of the reference to the shared latent codes of the unlabeled query (v). The latent codes are colored by cell-type annotations. **c** Batch effect correction enabled by UnitedNet. The UMAP latent space visualizations show DNA accessibility (i), RNA (ii), and shared latent codes (iii) as in (**b**). The latent codes are colored by different batches. **d** Boxplots comparing the performance of UnitedNet and scANVI on joint

group identification of the multiome ATAC + gene expression BMMCs dataset. Note that the performance of UnitedNet with supervised group identification task only and UnitedNet with cross-modal prediction task only are included as ablation studies. scANVI[42] is used for single-modality annotation transfer based on the ATAC or the gene expression modality. **e, f** Quantitative prediction results of UnitedNet and other ablation studies on the multiome ATAC + gene expression dataset. The prediction from DNA accessibility to RNA (**e**) and the prediction from RNA to DNA accessibility (**f**) are evaluated by the coefficient of determination and the area under the curve (AUC), respectively. Box, 75% and 25% quantiles. Line, median. Whisker, the maxima/minima or to the median $\pm 1.5\times$ inter quartile range (IQR). $n = 13$ folds cross validation for panels (**d–f**). **g** Heatmap comparing cross-modal predicted gene expression and DNA accessibility features with the ground truth. The values are averaged, and min-max scaled over annotated cell types. Source data are provided as a Source data file. The drawings in panel **a** were created with BioRender.com.

Next, we used ablation studies to validate whether multi-task learning can still achieve better performance in a supervised setting. We compared UnitedNet with a state-of-the-art annotation transfer method[42] and an ablated version of UnitedNet that only involved one task. We found that UnitedNet trained with supervised group identification and cross-modal prediction showed similar or better accuracy in both tasks (Fig. 5d–f). Finally, we evaluated whether the cross-modal prediction by UnitedNet could reconstruct cell-type-specific feature patterns by comparing the gene expression and DNA accessibility patterns across cell types between the ground truth and predicted results. We found a high degree of similarity in the gene expression-to-DNA accessibility and DNA accessibility-to-gene expression prediction, indicating the good performance of the cross-modal prediction task (Fig. 5g).

## UnitedNet indicates the relevance relationship between gene expression and DNA accessibility with cell-type specificity

We further dissected the UnitedNet trained by the multiome ATAC + gene expression data for cell-type-specific, cross-modal feature-to-feature relevance (Fig. 6 and Supplementary Fig. 14). We examined the effectiveness and robustness of our explainable learning in the similar way as discussed above (See "Methods"). These results support the effectiveness and robustness of our method for dissecting the trained UnitedNet with SHAP (Supplementary Figs. 12 and 15).

We then explored the biological values of the explainable learning in UnitedNet for the multiome ATAC + gene expression dataset. Using CD8+ T major cell type (both CD8+ T and CD8+ T naive cells) as an example (Fig. 6a), our results first identified a subset of genes (e.g., *CD8A*, *A2M*, *LEF1*, and *NELL2*) and DNA accessibility sites (e.g., *CD8A*, *DPP8*, *KDM2B*, and *KDM6B*) that were differentially expressed for the CD8+ T major cell type (Fig. 6b, d, e). Within these genes and DNA accessibility sites, the DNA accessibility sites *PROS1*, *KDM2B*, and *KDM6B* exhibited stronger relevance to CD8+ T cell-specific genes, suggesting their critical roles in CD8+ T cell functions (Fig. 6c). The results align with the previous studies that the elevated expression level of *PROS1* in the CD8+ T cell is a crucial regulatory signal that prevents overactive immune responses[43]. Meanwhile, the lack of *KDM2B* expression initiates T-cell leukemogenesis[44]. Notably, a recent study finds that *KDM6B*, a member of the same gene family as *KDM2B*, directly regulates the generation of CD8+ T cells by inducing DNA accessibility in effector-associated genes[45]. Our results further suggest that *KDM2B* may also potentially play an important role in regulating the generation of CD8+ T cells.

## UnitedNet utilizes spatial information in spatial-omics data and achieves high tissue region identification accuracy

Spatial omics is another important multimodal technology that allows for the measurement of spatially resolved multi-omics information in intact tissues[46–48]. However, spatial information is often not fully utilized

when analyzing spatial omics data for group identification tasks. UnitedNet is flexible to integrate different modalities as inputs, including spatial information. As demonstrated below, UnitedNet can utilize cell niche information (neighborhood gene expression information of each cell) as an additional modality to identify biologically meaningful groups and potentiate cross-modal prediction (see "Methods", Fig. 7a).

We first applied UnitedNet to a single-batch DBiT-seq embryo dataset, which simultaneously mapped the whole transcriptome and 22 proteins on an embryonic tissue[31]. Specifically, we generated the weighted average of RNA expression in the cell niche, which encodes spatial information, as the third modality for analysis. UnitedNet then combined gene expression, protein, and niche modalities for unsupervised joint identification of tissue regions and cross-modal prediction between gene expression and proteins. We benchmarked the accuracy of tissue region identification by considering the anatomic annotation of the tissue region from the original report as the ground truth[31]. UnitedNet achieved a higher unsupervised group identification accuracy compared with state-of-the-art methods (Fig. 7b). In addition, UnitedNet enabled the possibility of spatially resolved cross-modal prediction between several representative genes and protein expressions (Fig. 7c).

Next, we applied UnitedNet to an annotated multi-batch spatial-omics dataset for simultaneous supervised joint group identification and cross-modal prediction. We used a human dorsolateral prefrontal cortex (DLPFC) dataset that spatially maps gene expression and H&E staining on DLPFC brain slices from 12 batches[30]. Similarly, we used the gene expression, H&E staining-based morphological features, and cell niche modalities as inputs to UnitedNet. UnitedNet can successfully annotate the unseen DLPFC slice (Supplementary Fig. 16) and achieve higher or comparable accuracy than the other benchmarking methods and an ablated version of UnitedNet that did not use the alternating training scheme or the cell niche modality (Fig. 7d and Supplementary Fig. 17). We also visualized learned modality-specific weights and found that the weights of gene expression were higher than other modalities in the spatial DLPFC dataset, which aligns with the previous conclusion that gene expression is a more informative modality in the DLPFC dataset[30].

In addition, we explored whether UnitedNet can reduce the batch effect in the spatial DLPFC dataset, in a similar way as the analyses for the multiome ATAC + gene expression BMMC dataset. The results showed that UnitedNet maintained both good separability in the latent space and the ability to reduce the batch effect, enabling the higher or comparable performance in the group identification task compared to other ablation studies (Supplementary Fig. 18). Furthermore, UnitedNet enabled the possibility of cross-modal prediction between several representative genes and the H&E morphological features (Fig. 7e).

In summary, UnitedNet can extract spatial information as an input modality to enable both supervised and unsupervised group identification and cross-modal prediction for spatial-omics data.

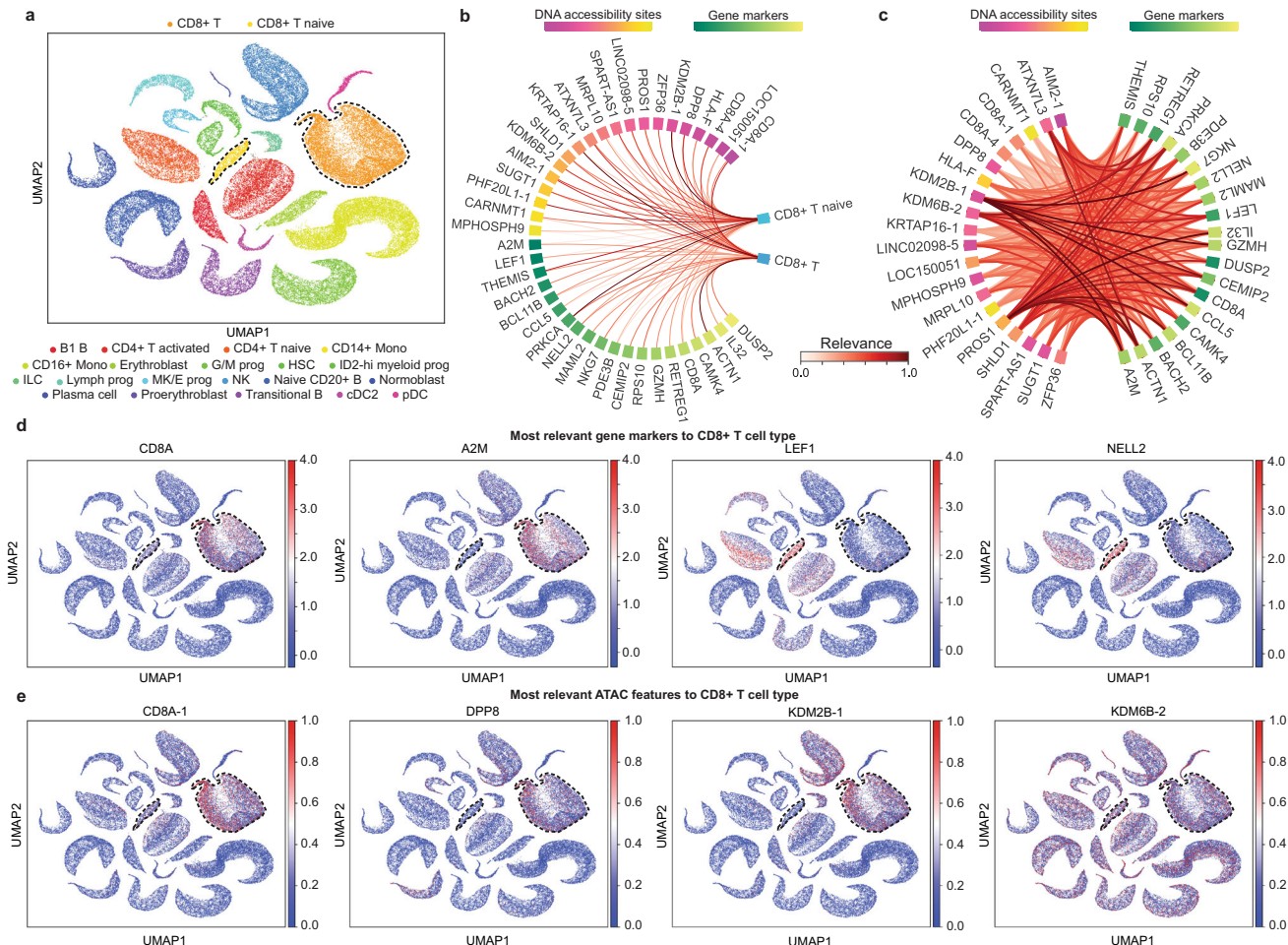

**Fig. 6 | Explainable learning for UnitedNet trained with multiome ATAC + gene expression data. a** UMAP representations of the shared codes of multiome ATAC + gene expression data in the shared latent space that are color-coded by the cell types identified by UnitedNet. The dashed line indicates the location of the CD8+ T and CD8+ T naive cells in the latent space. **b** Chord diagram showing the feature-to-group relevance for CD8+ T cells. The relevance values are min-max normalized to 0-1. **c** Chord diagram showing the DNA accessibility-to-gene expression relevance. The features used for cross-modal relevance analysis are from (**b**). The relevance values are min-max normalized to 0-1. **d, e** UMAP representations of the shared codes of multiome ATAC + gene expression data in the shared latent space are color-coded by most predicted cell types (**c**), most CD8+ T cells-relevant gene expression value (**d**), and DNA accessibility sites value (**e**). The dashed line indicates the location of the CD8+ T and CD8+ T naive cells in the latent space. Source data are provided as a Source data file.

## Discussion

We have demonstrated that UnitedNet can effectively integrate multiple tasks, such as the joint group identification and cross-modal prediction tasks and enable cross-modal relevance discovery through explainable multi-task learning for multi-modality data analysis. Through extensive ablation and benchmarking studies, we have validated that multi-task learning can achieve similar and better performance than single-task learning, single-modality analysis, and other state-of-the-art methods in both unsupervised and supervised settings. UnitedNet is applicable to a wide range of single-cell multi-modality biological datasets, including but not limited to multi-modality simulation data, multi-sensing data, multi-omics data, and spatial omics data. Moreover, the trained UnitedNet, which integrates multi-modal group identification and cross-modal prediction information, can be dissected by post hoc explainable learning methods to potentially uncover biological insights such as cell-type-specific, cross-modal feature-to-feature relevance from multi-modality biological data. The success of UnitedNet in analyzing multi-modality biological data will expand our ability to chart and predict cell states via combined multi-modality information in heterogeneous biological systems.

We envision several directions where UnitedNet could be used for data-driven scientific discoveries. In the group identification task,

UnitedNet can adaptively and effectively integrate multiple modalities measured from the same cells, potentially improving current cell typing systems to discover cell types that may not be identified using single-modality methods and helping to infer cell-type-specific phenotypes and functions[49]. For the cross-modal prediction task, UnitedNet may be able to predict end-point modality (e.g., gene expression) from continuous measurements of other modalities (e.g., electrical activity and bioimaging[50]). The enhanced performance of UnitedNet could enable the construction of a reliable predictive model, allowing for the use of continuous prediction to generate biological insights. Last but not least, UnitedNet could be used to suggest the most likely relevance relationships from presented multi-modal data, providing useful guidance for downstream wet-lab validation and potentially reducing time-consuming experiments.

Although the UnitedNet performed well for different multi-modality datasets, a number of improvements may be explored in the future, such as automatic searching for the optimal network configuration (e.g., the number of neuron nodes in each layer) and hyper-parameter (e.g., the learning rate that controls the optimization level in each training iteration) for deep neural networks. Moreover, there are some remaining questions such as why multi-task learning can improve multi-modal data analysis[19], how to reduce randomness in the neural

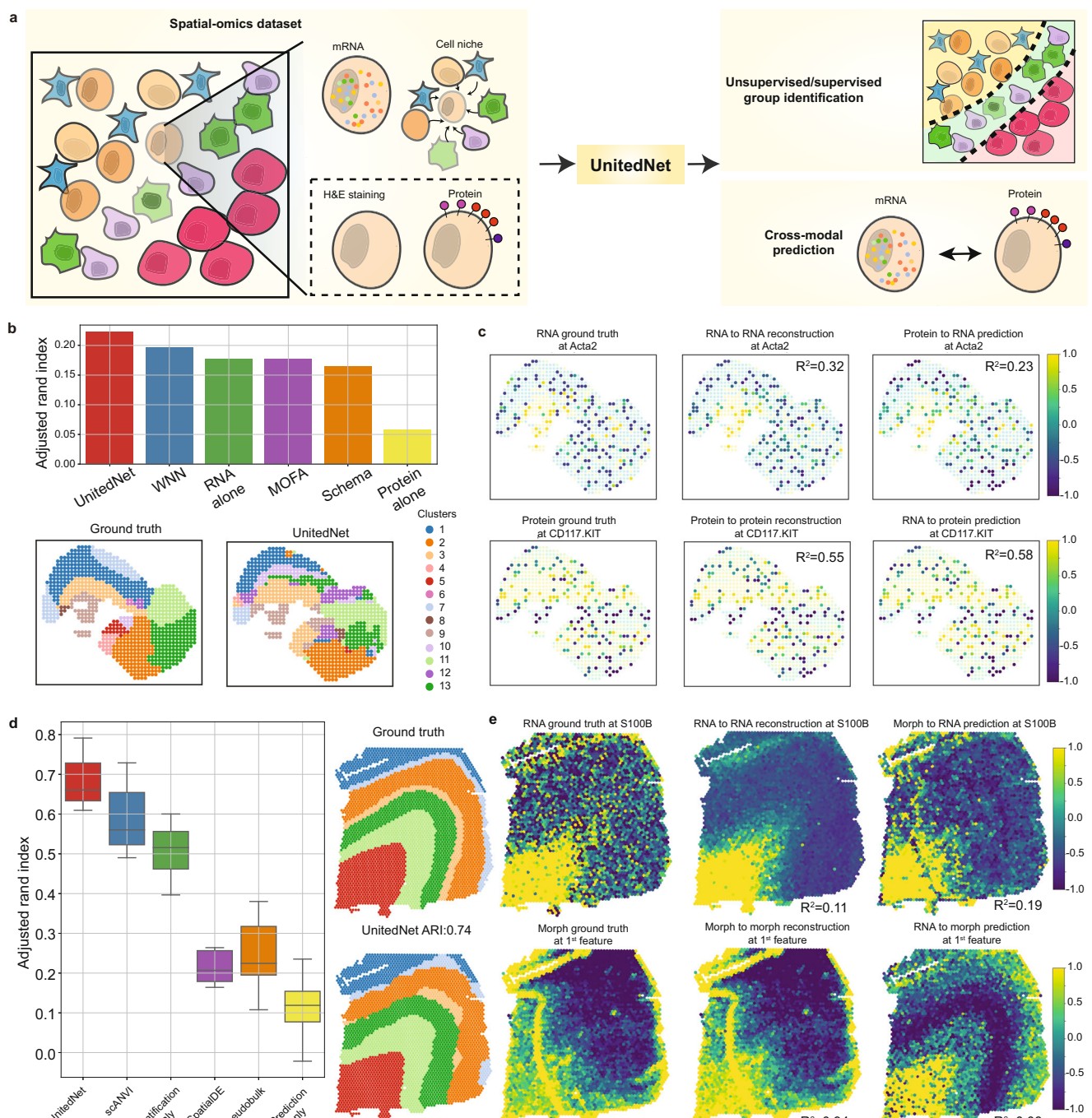

**Fig. 7 | Unsupervised group identification, joint annotation transfer, and cross-modal prediction of spatial-omics datasets by UnitedNet. a** Schematics of spatial-omics data analysis pipeline by UnitedNet. Spatial omics simultaneously measure spatially resolved multi-omics data in intact tissue networks. UnitedNet extracts the cell neighborhood information as an additional modality together with other modalities for unsupervised/supervised group identification and cross-modal prediction. **b** The performance comparison between UnitedNet and other methods for tissue region group identification on the DBiT-seq embryo dataset. **c** Results of cross-modal prediction between representative genes and proteins. **d** Boxplots comparing the performance of UnitedNet, scANVI, SpatialDE, and

Pseudobulk on tissue region identification of the DLPFC dataset. Note that SpatialDE and Pseudobulk results are from the original DLPFC paper[30]. Performance of UnitedNet trained with supervised group identification task only and trained with cross-modal prediction task only is included as ablation studies. scANVI is used for single-modality annotation transfer based on the gene expression modality. Box, 75% and 25% quantiles. Line, median. Whisker, the maxima/minima or to the median $\pm 1.5\times$ IQR. $n = 12$ folds cross validation. **e** Prediction results of representative genes, and morphological features from the DLPFC dataset. Source data are provided as a Source data file.

network training, how to design other loss functions to integrate more tasks (e.g., single-cell trajectory inferencing), and how to design wet lab experiments to validate the indicated cross-modal relevance. We envision that future mechanistic studies, including ablation experiments and theoretical analysis, could help address these questions.

## Methods
### UnitedNet
UnitedNet features both cross-modality prediction and joint group identification. The above two tasks are accomplished based on learned joint low-dimensional representations of different modalities, which

contain the essential information of the cells (Fig. 1b). Suppose that there are $V$ different modalities. Let $n$ denote the number of cells and $p^{(v)}$ denote the number of features in the $v$ th modality. Let $\mathbf{X}^{(v)} \in R^{n \times p^{(v)}}$ $(v = 1, \dots, V)$ denote the data from modality $v$ where its $i$ th row $\mathbf{x}_i^{(v)}$ corresponds to the cell $i$. For the prediction from modality $v_1$ to $v_2$, UnitedNet predicts $\mathbf{x}_i^{(v_2)}$ from the latent code (low-dimensional representation) obtained from $\mathbf{x}_i^{(v_1)}$. For group identification, UnitedNet first fuses the latent codes of $\mathbf{x}_i^{(1)}, \dots, \mathbf{x}_i^{(V)}$. Next, based on the shared latent codes, it returns a group index $k_i \in \{1, \dots, K\}$.

UnitedNet consists of encoders, decoders, discriminators, and a group identification module. It is trained based on a within-modality prediction loss, a cross-modality prediction loss, a generator loss, a discriminator loss, a contrastive loss, a clustering loss (for unsupervised group identification), and a classification loss (for supervised group identification). The details about the components and losses of UnitedNet are as follows.

## Encoders

For each modality $v = 1, \dots, V$, UnitedNet has one encoder $\text{Enc}^{(v)}(\cdot)$ that maps the features of each cell $i$ $(i = 1, \dots, n)$ to a modality-specific latent code $\mathbf{z}_i^{(v)}$ containing the most essential information of the data:

$$\mathbf{z}_i^{(v)} = \text{Enc}^{(v)}\left(\mathbf{x}_i^{(v)}\right). \tag{1}$$

The low-dimensional representations from different modalities are required to have the same number of components.

## Decoders

The decoder $\text{Dec}^{(v)}(\cdot)$ takes modality-specific latent codes as the input and maps them to the features of modality $v$. We denote the cross-modality predicted features from modality $v_1$ to $v_2$ (where $v_1 \neq v_2$) by

$$\widetilde{\mathbf{x}}_i^{(v_1,v_2)} = \text{Dec}^{(v_2)}\left(\mathbf{z}_i^{(v_1)}\right), \tag{2}$$

and the within-modality predicted features of modality $v$ by

$$\widetilde{\mathbf{x}}_i^{(v)} = \text{Dec}^{(v)}\left(\mathbf{z}_i^{(v)}\right). \tag{3}$$

## Discriminators

Discriminators assist the training of the generator that consists of the encoders and decoders. The discriminator $\text{Dis}^{(v)}(\cdot)$ of modality $v$ takes either the within-modality predicted features $\widetilde{\mathbf{x}}_i^{(v)}$ or original features $\mathbf{x}_i^{(v)}$ as the input and outputs a binary classification result, aiming to distinguish between $\widetilde{\mathbf{x}}_i^{(v)}$ and $\mathbf{x}_i^{(v)}$. The encoders and decoders improve their performance by increasing the error rate of the discriminators.

## Joint group identification module

Denote the number of groups to be $K$. The group identification module takes the modality-specific codes from all modalities $(\mathbf{z}_i^{(1)}, \dots, \mathbf{z}_i^{(V)})$ as the input and assigns it to one of the $K$ groups. It first fuses the data:

$$\mathbf{z}_i = \sum_{v=1}^{V} \eta_v \mathbf{z}_i^{(v)}, \tag{4}$$

where $\eta_1, \dots, \eta_V$ are nonnegative trainable weights with $\sum_{v=1}^{V} \eta_v = 1$. Next, the fused representation $z_i$ is passed through a fully connected layer: $\mathbf{h}_i = \text{layer}_1(\mathbf{z}_i)$. Then, to obtain the soft assignment of the group index, the intermediate output $\mathbf{h}_i$ is processed by another fully connected layer:

$$\boldsymbol{\alpha}_i = \text{layer}_2(\mathbf{h}_i) = \left(M_1(\mathbf{h}_i), \dots, M_K(\mathbf{h}_i)\right)^T, \tag{5}$$

where

$$M_k(\mathbf{h}) = \frac{\exp(W_k \mathbf{h})}{\sum_{t=1}^{K} \exp(W_t \mathbf{h})}, \tag{6}$$

and $W_k$ $(k = 1, \dots, K)$ is a vector of model coefficients. The group identification module assigns the group index $k_i = \arg\max_{k=1,\dots,K} M_k(\mathbf{h}_i)$ to cell $i$.

## Clustering loss

The clustering loss consists of three components. The first two components are adopted from the Deep Divergence-based Clustering (DDC)[32,33]. They ensure that the obtained groups are separable and compact. The third component is based on the self entropy[51]. It avoids trivial solutions where most of the cells are assigned to a small portion of the total number of groups.

**Component 1.** Component 1 reduces the two-by-two correlations between the cluster probability assignments (soft assignments) of different groups. It increases the group separability since those correlations are positively related to the similarity between different groups. Define the matrix $\mathbf{S} \in R^{n \times n}$ with matrix element $s_{i,j} = \exp\left(-||\mathbf{h}_i - \mathbf{h}_j||_2^2 / (2\sigma^2)\right)$, where $i = 1, \dots, n, j = 1, \dots, n$, and $\sigma$ is a hyperparameter. It measures the similarity between different cells. Denote $(M_k(\mathbf{h}_1), \dots, M_k(\mathbf{h}_n))^T$ by $\widetilde{\alpha}_k$, which is the soft assignments of the group $k$ for cells 1 to $n$. Then, the component 1 is calculated by

$$\mathcal{L}_{c1} = \binom{K}{2}^{-1} \sum_{k=1}^{K-1} \sum_{l>k}^{K} \frac{\widetilde{\alpha}_k^T \mathbf{S} \widetilde{\alpha}_l}{\sqrt{\widetilde{\alpha}_k^T \mathbf{S} \widetilde{\alpha}_k \widetilde{\alpha}_l^T \mathbf{S} \widetilde{\alpha}_l}}, \tag{7}$$

where $\widetilde{\alpha}_k^T \mathbf{S} \widetilde{\alpha}_l$ is an estimate of $\int M_k(\mathbf{h}) M_l(\mathbf{h}) d\mathbf{h}$, and it is minimized when $M_k(\mathbf{h})$ is orthogonal to $M_l(\mathbf{h})$. Accordingly, $\mathcal{L}_{c1}$ requires the values $M_k(\cdot)$ and $M_l(\cdot)$ $(k \neq l)$ to have low correlation.

**Component 2.** The component 2 pushes the soft assignment values of different groups to distinct corners of the simplexes in $R^K$, which also increases the group separability. Let $\mathbf{e}_k \in R^K$ denote a vector with its $k$ th element be one and other elements be zero. Therefore, $\mathbf{e}_k$ $(k = 1, \dots, K)$ is the $k$ th corner of the simplex. Recall that $\boldsymbol{\alpha}_i$ is the output from $\text{layer}_2$ in the group identification module. Let $\mathbf{m}_k \in R^n$ with its $i$ th element be $\exp\left(-||\boldsymbol{\alpha}_i - \mathbf{e}_k||_2^2\right)$, which measures the distance between the soft assignments $\boldsymbol{\alpha}_i$ and simplex corner $\mathbf{e}_k$. The component 2 is defined by

$$\mathcal{L}_{c2} = \binom{K}{2}^{-1} \sum_{k=1}^{K-1} \sum_{l>k}^{K} \frac{\mathbf{m}_k^T \mathbf{S} \mathbf{m}_l}{\sqrt{\mathbf{m}_k^T \mathbf{S} \mathbf{m}_k \mathbf{m}_l^T \mathbf{S} \mathbf{m}_l}}. \tag{8}$$

It enforces the orthogonality between $\exp\left(-||\boldsymbol{\alpha}_i - \mathbf{e}_k||_2^2\right)$ and $\exp\left(-||\boldsymbol{\alpha}_i - \mathbf{e}_l||_2^2\right)$. Therefore, the soft assignment output $(M_1(\mathbf{h}), \dots, M_K(\mathbf{h}))^T$ will tend to get close to one simplex corner (e.g., $\mathbf{e}_k$) instead of approaching multiple of them (e.g., both $\mathbf{e}_k$ and $\mathbf{e}_l$) at the same time. Consequently, the low-dimensional representation in the same group will be compact and those from different groups will be separated.

**Component 3.** Component 3 aims to avoid the trivial solution where most of the cells are assigned to a small proportion of the total groups. Let $\bar{\boldsymbol{\alpha}} \in R^K$ denote the averaged $\boldsymbol{\alpha}_i$ for $i = 1, \dots, n$, with its $k$ th element denoted by $\bar{\alpha}_k (k = 1, \dots, K)$. The third component is based on the negative entropy of $\bar{\boldsymbol{\alpha}}$:

$$\mathcal{L}_{c3} = \sum_{k=1}^{K} \bar{\alpha}_k \log \bar{\alpha}_k, \tag{9}$$

which aims to assign each group index with equal probability (i.e., $\bar{\alpha}_1 = , \cdots, = \bar{\alpha}_k = 1/K$). Therefore, it regularizes the groups by avoiding the assignment to be a few subsets of total clusters.

**Prediction loss**

The within-modality prediction loss is defined by

$$\mathcal{L}_{Wpredict} = \frac{1}{nV} \sum_{i=1}^{n} \sum_{v=1}^{V} \left\| \widetilde{\mathbf{x}}_i^{(v)} - \mathbf{x}_i^{(v)} \right\|_2, \tag{10}$$

which measures the distance between the within-modality predicted features $\widetilde{\mathbf{x}}_i^{(v)}$ and the original features $\mathbf{x}_i^{(v)}$. The cross-modality prediction loss is defined by

$$\mathcal{L}_{Cpredict} = \frac{1}{n\binom{v}{2}} \sum_{i=1}^{n} \sum_{v_1 < v_2} \left\| \widetilde{\mathbf{x}}_i^{(v_1,v_2)} - \mathbf{x}_i^{(v_2)} \right\|_2, \tag{11}$$

which measures the distance between the cross-modality predicted features from modality $v_1$ to $v_2$ and the original features from modality $v_2$. When the low-dimensional representation captures the essential information of the data, both $L_{Wpredict}$ and $L_{Cpredict}$ are expected to be small. When modality has binary observations (e.g., DNA accessibility), following a previous study[13], we replace equations 10 and 11 with binary cross-entropy loss.

**Generator loss and discriminator loss**

The generators and discriminators are trained by the least-squares loss[25]. We assign the within-modality predicted features $\widetilde{\mathbf{x}}_i^{(v)}$ with label one and the original features $\mathbf{x}_i^{(v)}$ with label zero. The generator loss is defined by

$$\mathcal{L}_{Gen} = \frac{1}{nV} \sum_{i=1}^{n} \sum_{v=1}^{V} \left\| \text{Dis}^{(v)}\left( \widetilde{\mathbf{x}}_i^{(v)} \right) - 1 \right\|_2^2, \tag{12}$$

which is minimized when $\text{Dis}^{(v)}\left( \widetilde{\mathbf{x}}_i^{(v)} \right) = 1$ for $i = 1, \ldots, n$ and $v = 1, \ldots, V$, namely, when the discriminator incorrectly classifies all the within-modality predicted features to one. The discriminator loss is defined by

$$\mathcal{L}_{Dis} = \frac{1}{nV} \sum_{i=1}^{n} \sum_{v=1}^{V} \left\| \text{Dis}^{(v)}\left( \widetilde{\mathbf{x}}_i^{(v)} \right) \right\|_2^2 + \frac{1}{nV} \sum_{i=1}^{n} \sum_{v=1}^{V} \left\| \text{Dis}^{(v)}\left( \mathbf{x}_i^{(v)} \right) - 1 \right\|_2^2, \tag{13}$$

which aims to make the discriminator classify the within-modality predicted features to zero and original features to one. This set of least-squares losses improves the quality of the trained generator, since it matches the essential goal of the generator, which is generating feature data with a distribution similar to that of the original features.

**Contrastive loss**

We apply the contrastive loss[32,34] to align the latent codes from different modalities. Define the cosine similarity by

$$s_{i,j}^{(v_1 v_2)} = \frac{\left(\mathbf{z}_i^{(v_1)}\right)^T \mathbf{z}_j^{(v_2)}}{\left\| \mathbf{z}_i^{(v_1)} \right\|_2 \cdot \left\| \mathbf{z}_j^{(v_2)} \right\|_2}, \tag{14}$$

where $i,j = 1, \ldots, n$. It is maximized when $\mathbf{z}_i^{(v_1)}$ and $\mathbf{z}_j^{(v_2)}$ are parallel. Let

$$l_i^{(v_1 v_2)} = -\log \frac{\exp\left( \frac{s_{i,i}^{(v_1 v_2)}}{\tau} \right)}{\sum_{s' \in \text{Neg}\left( \mathbf{z}_i^{(v_1)}, \mathbf{z}_i^{(v_2)} \right)} \exp\left( \frac{s'}{\tau} \right)}, \tag{15}$$

where $\text{Neg}\left( z_i^{(v_1)}, z_i^{(v_2)} \right)$ is obtained by sampling a fixed number of elements from the set $N_i = \{s_{i,j}^{(v_1 v_2)} : j = 1, \ldots, n, j \neq i, v_1, v_2 = 1, \ldots, V, \arg\max \alpha_i \neq \arg\max \alpha_j\}$, and $\tau$ is a hyperparameter. The term $\exp\left( s_{i,i}^{(v_1 v_2)}/\tau \right)$ aligns the modality-specific codes. For the denominator, the set $N_i$ contains all the cosine similarities between the latent codes of cell $i$ and those from the other cells that are grouped into different groups. The contrastive loss is defined by

$$\mathcal{L}_{Con} = \delta \cdot \frac{1}{n\binom{V}{2}} \sum_{i=1}^{n} \sum_{v_1 < v_2} l_i^{(v_1, v_2)}, \tag{16}$$

where $\delta$ is a hyperparameter.

**Classification loss**

Define vector $\mathbf{b}_i$ with its $k$ th element be one and the other $K-1$ elements be zero, where $k$ is the observed class label of cell $i$. Let $g_i = n_{k(i)}/n$ where $n_k$ ($k = 1,...,K$) is the number of cells with cell type label $k$. We assess the classification accuracy of the group identification module by the cross-entropy:

$$\mathcal{L}_{entropy} = -\frac{1}{n} \sum_{i=1}^{n} g_i \cdot \left( \log(M_1(\mathbf{h}_i)), \ldots, \log(M_K(\mathbf{h}_i)) \right)^T \mathbf{b}_i. \tag{17}$$

**Training procedure**

We first present the procedure where UnitedNet is trained without cell labels for cross-modality prediction and unsupervised group identification. It is trained iteratively with two steps: *group identification update step* and *prediction update step*. In the *group identification update step*, the encoder outputs the modality-specific codes and feeds them to the group identification module. Then, the group identification module fuses the modality-specific codes as shared latent codes and obtains the $K$-dimensional soft cluster assignments. The decoders output the within-modality predicted features $\widetilde{\mathbf{x}}_i^{(v)}$ based on the modality-specific codes. Next, the encoders and clustering module are updated by the group identification loss:

$$\mathcal{L}_{group} = \mathcal{L}_{c1} + \mathcal{L}_{c2} + \mathcal{L}_{c3} \tag{18}$$

The group identification step is repeated proportional to the modality number. In the *prediction update step*, the encoders output the low-dimensional representations from different modalities, feed them to the decoders, and obtain the cross-modality predicted features $\widetilde{\mathbf{x}}_i^{(v_1,v_2)}$ together with within-modality predicted features $\widetilde{\mathbf{x}}_i^{(v)}$. Then, the within-modality predicted features are fed into the discriminators. The discriminator is updated by the discriminator loss $\mathcal{L}_{Dis}$. Next, the encoder and decoder are updated by the sum of the within-modality prediction loss, cross-modality prediction loss, generator loss, and contrastive loss:

$$\mathcal{L}_{PGC} = \mathcal{L}_{Wpredict} + \mathcal{L}_{Cpredict} + \mathcal{L}_{Gen} + \mathcal{L}_{Con}. \tag{19}$$

The above two training steps are summarized in Algorithms 1 and 2, respectively. We acknowledge that the inherent stochasticity in the training of artificial neural networks, such as GPU computation, may result in subtle differences in the outcomes.

To train UnitedNet for supervised classification, we modify the group identification loss by:

$$\mathcal{L}_{group} = \mathcal{L}_{entropy}. \tag{20}$$

---

**Algorithm 1** Group identification update step

1: **for** $i = 1, \ldots, n$ **do**
2:   **for** $v = 1, \ldots, V$ **do**
3:     $\mathbf{z}_i^{(v)} = \mathrm{Enc}^{(v)}(\mathbf{x}_i^{(v)})$
4:     $\tilde{\mathbf{x}}_i^{(v)} = \mathrm{Dec}^{(v)}(\mathbf{z}_i^{(v)})$
5:   **end for**
6:   $\mathbf{z}_i = \sum_{t=1}^{v} \eta_v \mathbf{z}_i^{(v)}$
7:   $\mathbf{h}_i = \mathrm{layer}_1(\mathbf{z}_i)$
8:   $\boldsymbol{\alpha}_i = \mathrm{layer}_2(\mathbf{h}_i)$
9: **end for**
10: $\mathcal{L}_{group} = \mathcal{L}_{group}.Cal(\mathbf{h}_i, \boldsymbol{\alpha}_i, \tilde{\mathbf{x}}_i^{(v)})$ ▷ $i = 1, \ldots, n, v = 1, \ldots, V$
11: $\mathcal{L}_{group}.backward()$ ▷ Calculate the gradient
12: $step()$ ▷ Update the parameters

---

**Algorithm 2** Prediction update step

1: **for** $i = 1, \ldots, n$ **do**
2:   **for** $v = 1, \ldots, V$ **do**
3:     $\mathbf{z}_i^{(v)} = \mathrm{Enc}^{(v)}(\mathbf{x}_i^{(v)})$
4:     $\tilde{\mathbf{x}}_i^{(v)} = \mathrm{Dec}^{(v)}(\mathbf{z}_i^{(v)})$
5:     $\tilde{\mathbf{d}}_i^{(v)} = \mathrm{Dis}(\tilde{\mathbf{x}}_i^{(v)})$ ▷ Classify the reconstructed feature
6:     $\mathbf{d}_i^{(v)} = \mathrm{Dis}(\mathbf{x}_i^{(v)})$ ▷ Classify the original feature
7:     **for** $v' = 1, \ldots, v-1, v+1, \ldots, V$ **do**
8:       $\tilde{\mathbf{x}}_i^{(v,v')} = \mathrm{Dec}^{(v')}(\mathbf{z}_i^{(v)})$
9:     **end for**
10:   **end for**
11: **end for**
12: $\mathcal{L}_{Dis} = \mathcal{L}_{Dis}.Cal(\tilde{\mathbf{d}}_i^{(v)}, \mathbf{d}_i^{(v)})$ ▷ $i = 1, \ldots, n, v = 1, \ldots, V$. Calculate $\mathcal{L}_{Dis}$
13: $\mathcal{L}_{Dis}.backward()$ ▷ Calculate the gradient
14: step() ▷ Update the parameters
15: $\mathcal{L}_{PGC} = \mathcal{L}_{PGC}.Cal(\mathbf{x}_i^{(v)}, \tilde{\mathbf{x}}_i^{(v)}, \tilde{\mathbf{x}}_i^{(v_1,v_2)}, \tilde{\mathbf{d}}_i^{(v)}, \mathbf{z}_i^{(v)})$▷ $i = 1, \ldots, n, v = 1, \ldots, V$, $v_1 \neq v_2$, and $v_1, v_2 \in \{1, \ldots, V\}$.
16: $\mathcal{L}_{PGC}.backward()$ ▷ Calculate the gradient
17: $step()$ ▷ Update the parameters

---

**Explainable learning for feature relevance analysis using SHAP**

To provide insights into the importance of different features, we applied SHAP (SHapley Additive exPlanations)[35,52,53], which is commonly used to interpret machine learning models. The idea of this approach is to approximate the influence of a feature with respect to the output by a linear function while fixing the other features, and the function's coefficient corresponds to the Shapley value. The advantages of SHAP include its (1) theoretically based interpretability, (2) wide application scope, and (3) calculation procedure without the need to perturb the model or data, which is required by many other feature importance methods. We will first present the calculation procedure of the Shapley value and then explain about how it is applied to UnitedNet.

Suppose that we want to assess the importance of a feature $x_j$ for a function $f(\mathbf{x})$, where $\mathbf{x} = (x_1, \ldots, x_Q)^T$ and $j \in \{1, \ldots, Q\}$. Let $F$ denote the set of features in $\mathbf{x}$ and $S$ denote a subset of $F$. Let $|F|$ and $|S|$ denote the number of elements in the set $F$ and $S$, respectively. The Shapley value is calculated by

$$\phi_j(\mathbf{x}) = \sum_{S \subseteq F \setminus \{j\}} \frac{|S|!(|F| - |S| - 1)!}{|F|!} \left[ f_{S \cup \{j\}}(\mathbf{x}_{S \cup \{j\}}) - f_S(\mathbf{x}_S) \right] \quad (21)$$

where $F \setminus \{j\}$ stands for dropping the $j$ th feature from the set $F$, $\mathbf{x}_{S \cup \{j\}}$ is the elements from $\mathbf{x}$ whose feature variable belong to the union set $S \cup \{i\}$, $f_S(\mathbf{x}_S)$ is calculated by the taking the sample average of $f(\mathbf{x}_S, \mathbf{x}_{F \setminus S}^{(i)})$, $\mathbf{x}_{F \setminus S}^{(i)}$ is the $i$ th observation of features that are not in $S$, and $f_{S \cup \{j\}}(\mathbf{x}_{S \cup \{j\}})$ is calculated in a similar way for the set $S \cup \{j\}$. The Shapley value measures the importance of the $j$ th feature by calculating a weighted average of the changes in $f(\mathbf{x})$ after removing this feature.

For neural network $f(\mathbf{x}) = f^{(1)} \circ f^{(2)} \circ \cdots \circ f^{(L)}$, denote the dimension of the output of the $l$ th layer ($l = 1, \ldots, L$) by $L^{(l)}$, denote the $q$ th input feature of the $l$ th layer by $e_q^{(l)}$ and its sample average by $\bar{e}_q^{(l)}$. We may estimate Sharpley value of the above model in a computationally efficient way by Deep SHAP[35] with the following recursive formula:

$$\phi_{q,r}\left(f^{(l-1)} \circ f^{(l)}, \mathbf{e}^{(l-1)}\right) = \left(e_q^{(l-1)} - \bar{e}_q^{(l-1)}\right) \cdot \sum_{r^{(l-1)}=1}^{L^{(l-1)}} m^{(l-1)}\left(r^{(l-1)}, q\right) \cdot m^{(l)}\left(r, r^{(l-1)}\right),$$

$$(22)$$

where $\phi_{q,r}\left(f^{(l-1)} \circ f^{(l)}, \mathbf{e}^{(l-1)}\right)$ is the Sharpley value that measures the importance of $e_q^{(l-1)}$ with respect to the $r$ th element of the output of $f^{(l-1)} \circ f^{(l)}$,

$$m^{(l)}\left(r, r^{(l-1)}\right) = \phi_{r^{(l-1)}, r}\left(f^{(l)}, \mathbf{e}^{(l)}\right) / \left(e_{r^{(l-1)}}^{(l)} - \bar{e}_{r^{(l-1)}}^{(l)}\right), \quad (23)$$

$$m^{(l-1)}\left(r^{(l-1)}, q\right) = \phi_{q, r^{(l-1)}}\left(f^{(l-1)}, \mathbf{e}^{(l-1)}\right) / \left(e_q^{(l-1)} - \bar{e}_q^{(l-1)}\right). \quad (24)$$

Namely, we can compute the Sharpley value of $f^{(l-1)} \circ f^{(l)}$ by the Sharpley values of $f^{(l-1)}$ and $f^{(l)}$.

To identify features with high relevance to a specific group, for each cell, we calculated the Sharpley value of each input feature with respect to the soft assignment of that group. Next, all Sharpley values from the cells that are classified as this group are taken as absolute values and used to calculate an average value. The top $n$ features with the highest averaged values are interpreted as having high relevance with the group ($n = 7$ for the Patch-seq GABAergic neuron dataset and $n = 20$ for the multiome ATAC + Gene expression BMMCs dataset). To quantify the cross-modal feature-to-feature relevance within groups, we considered the high-relevance features of this group selected in the previous step. Next, we calculated the Sharpley value of each feature with respect to each other features from another modality. Then, the absolute values of the Sharpley values from different cells were aggregated by averaging. The features with relatively large values were viewed as important. The relevance relationship was visualized using a chord diagram with a python package *MNE-Connectivity*[54].

**Robustness test of explainable learning using cross-validations**

We tested the robustness of SHAP with respect to two sources of randomness in the explainable learning outcomes: the inherent randomness of the SHAP method and the randomness introduced by training the UnitedNet model on different data.

For the inherent randomness, we first trained UnitedNet on a fixed dataset to eliminate the randomness from model training. We then used $k$ folds cross-validation to the dataset to calculate the Shapley values from the trained UnitedNet. The calculation of Shapley values (using the Deep SHAP algorithm) requires a background dataset to determine the reference output of the model and calculation data to generate the Shapley value by comparing it to the background data[35]. In the $k$ folds cross-validation, we took each fold in turn as the calculation data and the remaining $k-1$ folds as the background data (resulting in $k$ cross-validation replications). We then counted the frequency of the top $n$ features identified by Shapley values in each cross-validation replication. If the top $n$ features are consistently identified across different cross-validation replications (with high frequency), we conclude that SHAP is robust with respect to its inherent randomness. For extensive data (e.g., the multiome ATAC + gene expression dataset) that require vast computing power, we used the average of each cell type as background data and a subset of data as the calculation data. We also tested the robustness of SHAP with respect to both its inherent randomness and the training of UnitedNet. The procedure is similar to the above one, except we retrained UnitedNet on the $k-1$ folds of the data.

## Evaluation metrics

The unsupervised and supervised group identification performance is evaluated by the adjusted rand index. The prediction performance is evaluated by the coefficient of determination and area under the ROC curve. The relationship of two tasks is evaluated by Pearson's correlation. All metrics are calculated by *scikit-learn*[55]. The details about these metrics are as follows.

### Adjusted rand index (ARI)

The adjusted rand index compares the clusters obtained from the model with the one from the cell type labels. Let $a_{k_1}(k_1 = 1, \ldots, K)$ denote the number of cells in the $k_1$ th cluster from the model and $b_{k_2}(k_2 = 1, \ldots, K)$ denote the number of cells in the $k_2$ th cluster from the observed cell type labels. Let $n_{k_1,k_2}$ $(k_1 = 1, \ldots, K, k_2 = 1, \ldots, K)$ denote the number of observations in both the $k_1$ th cluster from the model and $k_2$ th cluster from the cell type labels. The adjusted rand index is calculated by

$$\text{ARI} = \frac{\sum_{k_1,k_2} \binom{n_{k_1,k_2}}{2} - \left[\sum_{k_1} \binom{a_{k_1}}{2} \sum_{k_2} \binom{b_{k_2}}{2}\right] / \binom{n}{2}}{\frac{1}{2}\left[\sum_{k_1} \binom{a_{k_1}}{2} + \sum_{k_2} \binom{b_{k_2}}{2}\right] - \left[\sum_{k_1} \binom{a_{k_1}}{2} \sum_{k_2} \binom{b_{k_2}}{2}\right] / \binom{n}{2}}.$$
(25)

ARI is close to one when the clustering result from the model is close to the one from the observed cell type labels and close to zero for a random guess.

### Coefficient of determination ($R^2$)

Let $\mathbf{y}$ and $\widetilde{\mathbf{y}}$ denote the observed data and predicted data, respectively. Define $\bar{\mathbf{y}}$ to be the vector with the same length as $y$ and $\widetilde{y}$ and each of its elements equals the averaged value of $y$. The coefficient of determination is calculated by

$$R^2 = 1 - \frac{\|\widetilde{\mathbf{y}} - \mathbf{y}\|_2^2}{\|\bar{\mathbf{y}} - \mathbf{y}\|_2^2}.$$
(26)

It compares the mean square error (MSE) of the prediction $\widetilde{\mathbf{y}}$ from the model with the MSE of the baseline that takes the constant value $\bar{\mathbf{y}}$ as the prediction. It takes a value from $(-\infty, 1]$ and equals one when $\widetilde{\mathbf{y}} = \mathbf{y}$.

### Pearson's correlation

Let $\check{\mathbf{y}}$ denote a vector with the same length as $\widetilde{\mathbf{y}}$ and each of its elements equals the averaged value of $\widetilde{\mathbf{y}}$. The Pearson's correlation between $\mathbf{y}$ and $\widetilde{\mathbf{y}}$ is calculated by

$$r = \frac{\langle \widetilde{\mathbf{y}} - \check{\mathbf{y}}, \mathbf{y} - \bar{\mathbf{y}} \rangle}{\|\widetilde{\mathbf{y}} - \check{\mathbf{y}}\|_2 \cdot \|\mathbf{y} - \bar{\mathbf{y}}\|_2}.$$
(27)

It takes a value in $[-1, 1]$ and equals one or negative one when the prediction has positive or negative linear relationship with the observed data, respectively. It equals zero when the prediction and observed data have no linear relationship.

### Area under the curve (AUC)

For modeling the multiome ATAC + gene expression dataset, we have binarized the DNA accessibility data. Thus, to evaluate the prediction for those data, we adopt the area under the ROC curve, which was also used in the previous study[13]. Let $n_0$ and $n_1$ denote the number of zeros and ones from the observed data, respectively. Let $p_{0,i}$ $(i = 1, \ldots, n_0)$ and $p_{1,j}$ $(j = 1, \ldots, n_1)$ denote the model predictions for the two groups of observations, respectively. The area under the ROC curve is calculated by:

$$\text{AUC} = \frac{\sum_{i=1}^{n_0} \sum_{j=1}^{n_1} I_{\{p_{0,i} < p_{1,j}\}}}{n_0 \cdot n_1},$$
(28)

where

$$I_{\{p_{0,i} < p_{1,j}\}} = \begin{cases} 1, & \text{when } p_{0,i} < p_{1,j}, \\ 0, & \text{otherwise}. \end{cases}$$
(29)

AUC takes values between zero and one, and one corresponds to a perfect prediction.

## Dataset used for multi-task learning in UnitedNet

The input of UnitedNet can be various multi-modality dataset. We conclude that four major categories of such datasets include multi-modal simulation datasets, multi-modal sensing datasets, multi-omics datasets, and spatial omics datasets. The experimental details are specified in the following sections.

## Multi-modality datasets used for the demonstration of UnitedNet

**Dyngen simulated dataset.** We use *Dyngen*[26] to simulate the four-modality dataset. Specifically, we generate 500 cells with simulated DNA, pre-mRNA, mRNA, and protein modalities, each modality containing 100 dimensions of features. Meanwhile, the ground truth cell-type annotations are generated along with the dataset. For the parameter of the Dyngen simulator, we use the default setting of a linear backbone model in the tutorial of Dyngen with the functions including *backbone_linear, initialize_model*, and *generate_dataset*.

**MUSE simulated dataset.** We apply the simulator in *MUSE*[27] to simulate two-modality inputs to assess the robustness of UnitedNet with one low-quality modality. We simulate 11 two-modality datasets with 1000 cells and 10 cell types. Each modality contains 500 modality-specific features. For each of the 11 datasets, one of the modalities is simulated with a controllable decay coefficient. We use 0.01, 0.1, 0.2, 0.3, 0.4, 0.5, 0.6, 0.7, 0.8, 0.9, and 1 as different decay coefficients when benchmarking with other methods.

**Patch-seq GABAergic neuron dataset.** We use a Patch-seq dataset that simultaneously characterizes the morphological (M), electrophysiological (E), and transcriptomic (T) features obtained from GABAergic interneurons in the mouse visual cortex[29]. We use the same dataset after quality control of previous research[11], in which 3395 neurons remain for E-T analysis and 448 neurons remain for M-E-T analysis. We standardize the input matrices of each modality to make the mean value and the standard deviation of all features in each cell to be 0 and 1, respectively.

**Multiome ATAC + gene expression BMMCs dataset.** We use a multiome ATAC + gene expression dataset that simultaneously combines gene expression and genome-wide DNA accessibility obtained in BMMC tissue from 10 donors and 4 tissue sites[7]. In addition to the quality control in the previous study, we use the standard preprocessing procedure for the multiome ATAC + gene expression BMMCs datasets[6]. For the preprocessing of the gene expression modality, we use median normalization and the *log1p* transform and standardization and select the top 4000 most variable genes through *Scanpy*[56]. For the preprocessing of the DNA accessibility modality, we binarize the data by replacing all nonzero values with a value of 1 and select the top 13,634 most variable DNA accessibility features through *Scanpy*. We use the *ChIPseeker*[57] and *scanpy.var_names_make_unique* to annotate the DNA-accessibility peaks.

## UnitedNet on spatial omics datasets

**Generating the niche expression modality.** Using the measured expression of RNAs of cells or spots, we incorporate the spatial information of each cell or spot and generate a weighted average expression of RNAs. With two-dimensional spatial coordinates $(s_i^1, s_i^2)$ and modality $v$ with its $i$ th row $x_i^{(v)}$ corresponding to the cell/spot $i$, we compute the niche modality for modality $v$ that is denoted by $x^{(v\ niche)}$ with $(v = 1, \ldots, V)$. For cell/spot $i$, we compute $x_i^{(v\ niche)}$ by:

$$x_i^{(v\ niche)} = \sum_{j=1}^{J} x_j^{(v)} \cdot w_{ij}, \tag{30}$$

where $j \in \{1, \ldots, J\}$ denotes cells/spots that belong to the J-nearest neighbors of cell/spot $i$, and $w_{ij}$ is calculated by:

$$w_{ij} = \frac{1/distance\left\{\left(s_i^1, s_i^2\right), \left(s_j^1, s_j^2\right)\right\}}{\sum_{j=1}^{J} 1/distance\left\{\left(s_i^1, s_i^2\right), \left(s_j^1, s_j^2\right)\right\}}, \tag{31}$$

where $distance\{\cdot\}$ denotes the Euclidean distance between two vectors.

**UnitedNet on DBiT-seq embryo dataset.** We use the DBiT-seq embryo dataset[31], where the following three modalities of 936 spots in DBiT-seq are taken: mRNA expression, protein expression, and niche mRNA expression. For modality of mRNA expression, we normalize the raw count matrix using function *scanpy.pp.normalize_total* from *scanpy* and select the top 568 differentially expressed genes. For the modality of protein expression, we normalize the raw count matrix and used 22 kinds of proteins. The niche modalities are generated based on normalized mRNA expression. For the first task of tissue region characterization, ground truth tissue region labels are extracted from the original research[31] which is the anatomic annotation of major tissue regions based on the H&E image. We compare clustering results from UnitedNet with those from other state-of-the-art methods. We validate their performance by adjusted rand index. For the parallel task of prediction across modalities, although three modalities are used as inputs to the UnitedNet model, we focus on prediction between the first and second modalities: mRNA expression and protein expression. Since there is only one batch in DBiT-seq public datasets, we split the total 936 spots in the DBiT-seq embryo dataset into the training dataset (80%, 748 spots) and testing dataset (20%, 188 spots) for the prediction task.

**UnitedNet on DLPFC dataset.** We use the human adult dorsolateral prefrontal cortex (DLPFC) datasets with 12 batches[30]. We use the following three modalities: mRNA expression, morphological features extracted from the H&E staining images, and niche mRNA. For modality of mRNA expression, we normalize the raw count matrix and select the top 2365 differentially expressed genes. We use a pre-trained convolutional neural network[58] to extract morphological features from the H&E staining images implemented by *stLearn*[59]. A 50-dimensional morphological feature is used as the second modality for each spot. For the supervised group identification task, we use 11 batches with their tissue region annotations to train a UnitedNet model. Then we apply the trained model to the remaining batch to identify tissue region annotation and perform the cross-modal prediction between the H&E image features and mRNA expression. We compare the identification performance of UnitedNet with *SpatialDE PCA* and *pseudobulk PCA* from the original DLPFC paper[30]. After the identification task, we applied a refinement step for the clustering result following SpaGCN with a number of 35 spots in the nearest neighbors[60].

## Statistics and reproducibility

In total, experiments and analyses were conducted on 7 different publicly available multi-modality datasets. No statistical methods were used to pre-determine dataset number and sizes, but they are similar to those reported in previous publications[9,16,27,48]. No data were excluded from the analyses. Statistical comparisons were performed using Python 3.7 and Scipy 1.7.3 with appropriate inferential methods, as indicated in the figure legends. Graphs were created using Python 3.7, matplotlib 3.5.1, and seaborn 0.11.2. Statistical results in the figures are presented as exact *P* value. Cross-validation was used to verify the performance on each dataset. Each fold of the cross validation was randomly splitted from the dataset or used as the existing biological group identities. The conclusions were drawn from the analysis of multiple experiments. Blinding was not relevant to the multi-modal data analyses because the datasets were not divided into control and experimental groups.

## Reporting summary

Further information on research design is available in the Nature Portfolio Reporting Summary linked to this article.

## Data availability

The Dyngen simulation data used in this study are available in the https://github.com/dynverse/dyngen. The MUSE simulation data used in this study are available in the https://github.com/AltschulerWu-Lab/MUSE. The Patch-seq GABAergic neuron dataset used in this study are available in the https://github.com/AllenInstitute/coupledAE-patchseq and https://knowledge.brain-map.org/data/1HEYEW7GMUKWIQW37BO/collections. The multiome ATAC + gene expression BMMCs dataset used in this study is available in the GEO database under accession code "GSE194122". The DBiT-dataset used in this study is available in the GEO database under accession code "GSE137986". The DLPFC dataset used in this study is available in the OpenNeuro database under accession code "ds002076". All processed data used in this manuscript have been deposited in Zenodo database under accession code "7708592". All other relevant data supporting the key findings of this study are available within the article and its Supplementary Information files or from the corresponding author upon reasonable request. Source data are provided with this paper.

## Code availability

Source code and demonstration code are made available at https://github.com/LiuLab-Bioelectronics-Harvard/UnitedNet and https://doi.org/10.5281/zenodo.7708592[61].

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

## Acknowledgements

We thank Jane Salant for her helpful comments on the manuscript. J.L., J.D., and N.L. acknowledge the support from the NSF ECCS-2038603. J.L. acknowledges the support from NIH/NIDDK 1DP1DK130673 and William F. Milton Fund. J.D. acknowledges the support from the Army Research Laboratory and the Army Research Office under grant number W911NF-20-1-0222. Y.H. acknowledges the support from the James Mills Peirce Fellowship from the Graduate School of Arts and Sciences of Harvard University. Schematics in Figs. 1a, 3a, and 5a were partially created with BioRender.com.

## Author contributions

J.L., J.D., and X.T. conceived the idea and designed the research. X.T. designed and developed the model. J.Z. and Z.L. provided critical discussions during the development. X.T., Y.H. and J.Z. conducted analyses on simulated and biological datasets. J.L., J.D., X.T., J.Z., Y.H., X.Z., Z.L., S.P., E.B.H., Z.R., H.S., Y.Y., X.W., and N.L. prepared figures and wrote the manuscript. J.L. and J.D. supervised the study.

## Competing interests

The authors declare no competing interests.
