## [Peer Review File · Nature Communications]

REVIEWER COMMENTS

Reviewer #1 (Remarks to the Author):

Tang et al. describe a framework, UnitedNet, for integrating single-cell multimodal analysis tasks using a multi-task deep neural network. The authors demonstrate the effectiveness of their multi-task approach first on simulated multimodal data; they perform a series of ablation studies, which show that the components of their neural network architecture design, specifically the use of multi-task learning and discriminator networks, are critical to UnitedNet's performance on joint group identification and cross-modal prediction tasks. The authors also showcase UnitedNet's ability to perform these tasks on several real single-cell multimodal datasets that feature a broad range of different modalities. On these datasets, they present UnitedNet's ability to perform annotation transfer and separately use neural interpretability methods to prioritize cell type-specific gene regulatory patterns learned by the neural network model. Overall, this work presents a compact framework that uses practical techniques in machine learning to improve upon the specific single-cell multimodal tasks of group identification and cross-modal predictions.

While very nice work, more comprehensive benchmarking is needed to better support the authors' claims of UnitedNet's superior performance relative to other state-of-the-art methods. A more thorough exploration of the utility of this framework in uncovering novel biological insights would also improve the impact of this work. The authors do use neural interpretability methods to recapitulate some past findings in a limited manner, but more detailed explanations or validation of these downstream results would be helpful in showcasing UnitedNet's potential applicability for discovering novel biology.

Major Comments

- The authors benchmark UnitedNet against several state-of-the-art multimodal integration and cross-modal prediction methods over the course of the paper, but do not consistently benchmark against the same set of methods for each prediction task. For example, MOFA is used in benchmarking the group identification task for the Patch-seq dataset, but not for the simulated DynGen dataset or other multimodal datasets for which UnitedNet's performance in the group identification task was also evaluated.
- How do the neural interpretability methods used on UnitedNet compare to another multimodal integration technique that has built-in interpretability capabilities: Schema (<https://genomebiology.biomedcentral.com/articles/10.1186/s13059-021-02313-2>)?
- Relatedly, how does UnitedNet compare against MOFA, totalVI, and Schema for unsupervised group identification? Some benchmarking has been done to compare UnitedNet against other methods for learning joint multimodal representations, but Schema has been shown to outperform several of these state-of-the-art methods in the unsupervised group identification task.
- The authors use SHAP to interpret trained UnitedNet models and to generate feature-cell type and feature-feature maps, but validation of these maps is currently limited to brief mentions of how certain aspects of these maps recapitulate a few prior biological findings. A more thorough and quantitative validation procedure would be helpful in building support for the effectiveness and utility of these maps (e.g., using the Shapley values as a predictor for marker genes and showing statistics for their predictability of these genes.) Similarly, how robust are the outputs of these neural interpretability methods to different model initializations or hyperparameter choices?
- Related to the previous point, details regarding how Shapley value thresholds are chosen to ascertain sufficient relevance in feature-cell type or feature-feature maps appear to be missing in the text.
- While UnitedNet demonstrates superior performance in the group identification and cross-modal prediction tasks, the manuscript would benefit from a deeper exploration or discussion of the utility of these specific tasks in generating new biological insights. It is presently not made clear in the text how one can leverage UnitedNet's performance gains on these tasks to contribute to discoveries outside of

applying neural interpretability methods to dissect the trained model.

Minor Comments

- There are a few points in the main text and figure captions where ATAC-seq is described as measuring both gene expression and chromatin accessibility in the same cell, but ATAC-seq refers only to the chromatin accessibility modality. This led to confusion about the experiments performed for the BBMCs dataset, which seem to profile both gene expression and chromatin accessibility modalities.
- The authors claim that UnitedNet's unsupervised group classification predictions display a high degree of congruence with previously reported results. Heatmaps are provided to qualitatively show this congruence, but no statistics are shown in the text to provide quantitative support for this claim. Also, how do these unsupervised classification results compare with other state-of-the-art unsupervised classification methods?
- Recent tools have been developed to infer feature-feature maps across single-cell modalities (e.g., GrID-Net, ICLR 2022, for peak-gene links) and should be referenced alongside the use of neural interpretability methods for generating such maps.

Reviewer #2 (Remarks to the Author):

This work by Tang etc. reported a multi-modality analysis pipeline that can integrate data from different modalities and further explore the relationship between them. The UnitedNet was validated and demonstrated impressive performance with quite a few datasets with different modalities, including patch-seq, ATAC-seq, DBiT-seq, and spatial transcriptomics. This work is an important contribution and represents a significant advance to the multimodal data integration of high throughput genomics research, given the fact that more multiomics-techniques in single cell and spatial fields are developed.

The Github is not accessible for review at present. This has to be corrected in a revised submission and I expected to see the Github page when the revised version was received.

Various metrics were used throughout the manuscript to compare the performance, can the authors justify the reason for switching the metric between different datasets? Will one metric contradict another metric regarding performance or accuracy?

Is there a way to show the weight of each modality for the clustering analysis? Quite like what is done in the WNN integration.

The batch effects were indeed corrected with UnitedNet. But how this alternating training feature can reduce the batch effects is not clearly explained or analyzed.

Point-by-point Response (Nature Communications NCOMMS-22-34542-T)

We were delighted to see the strongly positive comments from all of the reviewers, e.g., “*this work presents a compact framework that uses practical techniques in machine learning to improve upon the specific single-cell multimodal tasks*”, “*very nice work*”, “*UnitedNet demonstrates superior performance in the group identification and cross-modal prediction tasks*”, “*The UnitedNet was validated and demonstrated impressive performance*”, and “*This work is an important contribution and represents a significant advance to the multimodal data integration of high throughput genomics research*”.

We are incredibly grateful to the reviewers and Editor for their close read and helpful suggestions. We have taken care to address each suggestion in full and believe that these revisions have further strengthened the manuscript. Thank you for your careful attention and for helping to improve the quality of our work.

Response to Reviewer #1 Comments:

General Comment: Tang et al. describe a framework, UnitedNet, for integrating single-cell multimodal analysis tasks using a multi-task deep neural network. The authors demonstrate the effectiveness of their multi-task approach first on simulated multimodal data; they perform a series of ablation studies, which show that the components of their neural network architecture design, specifically the use of multi-task learning and discriminator networks, are critical to UnitedNet’s performance on joint group identification and cross-modal prediction tasks. The authors also showcase UnitedNet’s ability to perform these tasks on several real single-cell multimodal datasets that feature a broad range of different modalities. On these datasets, they present UnitedNet’s ability to perform annotation transfer and separately use neural interpretability methods to prioritize cell type-specific gene regulatory patterns learned by the neural network model. Overall, this work presents a compact framework that uses practical techniques in machine learning to improve upon the specific single-cell multimodal tasks of group identification and cross-modal predictions.

While very nice work, more comprehensive benchmarking is needed to better support the authors’ claims of UnitedNet’s superior performance relative to other state-of-the-art methods. A more thorough exploration of the utility of this framework in uncovering novel biological insights would also improve the impact of this work. The authors do use neural interpretability methods to recapitulate some past findings in a limited manner, but more detailed explanations or validation of these downstream results would be helpful in showcasing UnitedNet’s potential applicability for discovering novel biology.

Response to General comment: We would like to express our gratitude to the reviewer for the positive comments and constructive feedback on our work. We have made substantial revisions to the manuscript in response to reviewer’s suggestions, including additional performance benchmarking, effectiveness validation, robustness analyses, and further discussion to address each comment. Below is a summary of our responses to the reviewer's comments:

1. **Comprehensive benchmarking analyses to evaluate the performance of UnitedNet.** We have thoroughly compared the performance of UnitedNet to a range of *state-of-the-art* methods, including MOFA2, totalVI, Schema, and WNN on multiple datasets. The results demonstrate that UnitedNet consistently performs well in comparison with these methods.
2. **Quantitative validation of interpretable learning.** In response to the reviewer's suggestion, we have conducted a more quantitative and thorough validation of the interpretability of UnitedNet. The results show that our method can effectively and consistently identify the cell-type-specific features by dissecting the trained model. We have also discussed the differences in the interpretability between UnitedNet and Schema.
3. **Robustness test of interpretable learning.** We have demonstrated the robustness of our interpretable learning method, showing that it produces consistent results under different model initialization settings. Inspired by the reviewer's comment, we have further improved our method by using cross-validation to select the most reliable features for higher statistical confidence.
4. **Discussion of uncovering biological insights with UnitedNet.** We have elaborated on the potential of UnitedNet to combine an arbitrary number of modalities with an adaptive weight scheme for group identification and cross-modal prediction tasks. This capability can further enhance the performance of tasks such as discovering new cell types, predicting unmeasurable modalities, and integrating spatial information as a new modality for spatial multimodal data analysis in biology. We believe that these capabilities, in addition to its multimodal data interpretability, make UnitedNet a valuable tool for uncovering insights in biological research.

We greatly appreciate the valuable suggestions provided by the reviewer, which have further strengthened our manuscript.

Comment #1: The authors benchmark UnitedNet against several *state-of-the-art* multimodal integration and cross-modal prediction methods over the course of the paper, but do not consistently benchmark against the same set of methods for each prediction task. For example, MOFA is used in benchmarking the group identification task for the Patch-seq dataset, but not for the simulated DynGen dataset or other multimodal datasets for which UnitedNet's performance in the group identification task was also evaluated.

Response to Comment #1: We thank the reviewer for the suggestion. In our revised manuscript, we consistently benchmark UnitedNet against the same set of *state-of-the-art* methods for each task.

First, in the *supervised group identification task*, we compared the performance of UnitedNet and scANVI¹ for annotation transfer on the multiome ATAC+gene expression² and spatial DLPCF dataset as scANVI has been shown to be the *state-of-the-art* method for annotation transfer. The results demonstrated that UnitedNet outperformed scANVI in annotation transfer task using either

gene expression or ATAC modalities (**Response Figure 1-1a**). In addition, for the spatial multimodal biological datasets, SpatialDE³ and Pseudobulk⁴ were designed for the tissue region identification task, which can also be considered a supervised group identification task. Therefore, we compared the performance of UnitedNet with SpatialDE and Pseudobulk for DLPFC dataset. The results showed that UnitedNet outperformed SpatialDE and Pseudobulk (**Response Figure 1-1b**). Notably, as SpatialDE and Pseudobulk were not designed for multiome ATAC+gene expression datasets, we did not compare the performance of these two methods on the multiome ATAC+gene expression dataset.

Second, in the *unsupervised group identification task*, we conducted multimodal cell clustering on DynGen simulated datasets⁵, DBiT-seq⁶, and Patch-seq⁷ datasets. As suggested by the reviewer, we compared the performance of UnitedNet with four other *state-of-the-art* methods, including MOFA⁸, totalVI⁹, Schema¹⁰, and WNN¹¹, as well as Leiden clustering¹², which uses single modality, on DynGen simulated datasets, DBiT-seq, and Patch-seq datasets (shown in **Response Figure 1-2**). The results showed that UnitedNet consistently performed similarly or better in terms of group identification accuracy compared with the other methods, except for Schema, which performed better in the Patch-seq neuronal subtype group identification task.

In addition, we updated the results from Schema, WNN, MOFA for the latent space separability test for consistency. In this comparison, we used electrophysiology and morphology modalities as inputs to identify cell types. We examined the performance of this multimodal group identification task (cell typing) by comparing the cell typing results with transcriptomics-defined cell types as well as evaluated the latent space separability across all the methods. The results showed that UnitedNet consistently achieved the highest identification accuracy and latent space separability compared with the *state-of-the-art* methods (**Response Figure 1-2 d-j**). Note that due to totalVI's requirement for two-modality integer data as input⁹, we were unable to compare its performance on the Patch-seq data with non-integer electrophysiological and morphological features. These results further confirm the consistently good performance of UnitedNet when compared with *state-of-the-art* methods.

Finally, in the *cross-modal prediction task*, we conducted benchmarking experiments on the Patch-seq and DBiT-seq datasets. We compared the performance of UnitedNet with the coupled autoencoder on the Patch-seq datasets. The coupled autoencoder is the *state-of-the-art* method for cross-modal prediction between gene expression and electrophysiological features in the Patch-seq dataset^{13,14}. The results showed that UnitedNet outperformed the coupled autoencoder when predicting electrophysiology from gene expression and vice versa. However, the coupled autoencoder was not designed for spatial biological data analysis. Therefore, we did not further apply it to the DBiT-seq datasets. To our knowledge, there is no other method specifically designed for the cross-modal prediction task for spatial biological datasets like the DBiT-seq datasets.

Response Figure 1-1. A consistent benchmarking against the *state-of-the-art* methods in the supervised group identification task. **a, Boxplots comparing the performance of UnitedNet, scANVI, SpatialDE, and Pseudobulk on tissue region identification of the DLPFC dataset. Note that SpatialDE and Pseudobulk result is from the original DLPFC paper. Performance of UnitedNet trained with supervised group identification task only and trained with cross-modal prediction task only are included as ablation studies. scANVI is used for single-modality annotation transfer based on the gene expression modality. **b**, Boxplots comparing the performance of UnitedNet and scANVI on tissue region identification of the multiome ATAC+gene expression BMMC dataset. Note that the performance of UnitedNet with supervised group identification task only and UnitedNet with cross-modal prediction task only are included as ablation studies. scANVI is used for single-modality annotation transfer based on the DNA-accessibility (ATAC) or the gene expression modality.**

Response Figure 1-2. Comparison of the performance of UnitedNet with *state-of-the-art* methods in the unsupervised group identification task. a, Barplot showing performance comparison between UnitedNet and *state-of-the-art* methods including single-modality Leiden clustering, Schema, WNN, MOFA, and totalVI on the DynGen simulated dataset in the joint group identification task. The dashed lines represent the performance of joint group identification using Leiden clustering on the simulated modality of Pre-mRNA, Protein, DNA, and mRNA from DynGen dataset. **b**, Barplot showing performance comparison between UnitedNet and *state-of-the-*

art methods including single-modality Leiden clustering, Schema, WNN, and MOFA on the DBiT-seq embryo dataset in the joint group identification task. Note that totalVI was not included in the comparison because it cannot take the non-integer data (*i.e.*, niche RNA feature) as inputs. **c**, Barplot showing performance comparison between UnitedNet and *state-of-the-art* methods including single-modality Leiden clustering, Schema, WNN, and MOFA on the Patch-seq GABAergic neuron dataset using transcriptome, electrophysiology, and morphology modalities (MET) for MET cell type identification. The benchmarking was conducted on both major- and sub-MET type identification tasks. Note that totalVI was not included in the comparison because it cannot take the non-integer data (*i.e.*, electrophysiology feature and morphological features) as inputs. **d**, Barplot showing performance comparison between UnitedNet and *state-of-the-art* methods including single-modality Leiden clustering, Schema, WNN, and MOFA on the Patch-seq GABAergic neuron dataset using electrophysiology and morphology modalities for cell type identification. Note that totalVI was not included in the comparison because it cannot take the non-integer data (*i.e.*, electrophysiology feature and morphological features) as inputs. The separability of cell clusters defined by electrophysiology and morphology modalities in the latent space was evaluated against the transcriptomics-defined cell types and quantified using the Silhouette score. **e-j**, The latent space visualization by UMAP of UnitedNet (**e**), WNN (**f**), Schema (**g**), electrophysiology data alone (**h**), and morphology alone (**i**), and MOFA2 (**j**). The latent codes are colored to represent different cell type labels.

Revisions to the manuscript: the results of **Response Figure 1-1** and **Response Figure 1-2** have been updated in **Fig. 2**, **Fig. 3**, **Fig. 5**, **Fig. 7**, **Supplementary Fig. 4**.

We have included the following edits to the revised manuscript (New texts are highlighted bold): “*To evaluate the performance of UnitedNet, we used a simulated dataset containing four modalities (DNA, pre-mRNA, mRNA, and protein) with their ground truth labels from Dyngen, a multi-omics biological process simulator²⁷ (Fig. 2a and see Methods). We first benchmarked the unsupervised joint group identification performance of UnitedNet against the state-of-the-art multi-modal integration methods, including Schema¹⁷, Multi-Omic Factor Analysis (MOFA)³⁶, totalVI¹⁶, and Weighted Nearest Neighbor (WNN)⁹. We applied the Leiden clustering method to cluster the integrated joint representations from these methods and used single-modality clustering³⁷ using Leiden as the performance baseline. As a result, UnitedNet consistently exhibits similar or better unsupervised joint group identification accuracy compared with the single-modality Leiden clustering and other state-of-the-art methods (Fig. 2b and Methods).*”

“*We first benchmarked the unsupervised group identification performance of UnitedNet by combining cell electrophysiological and morphological features to identify transcriptomic cell types³⁸. Compared with other state-of-the-art cell typing methods including Schema¹⁷, Multi-MOFA³⁶, totalVI¹⁶, WNN⁹, and Leiden clustering using single modality data, the performance*

of cell typing results by UnitedNet demonstrates substantial improvement in terms of cell type separability and identification accuracy (Supplementary Fig. 4).”

“Next, we performed simultaneous unsupervised joint group identification analysis and cross-modal prediction on the morphological-electrophysiological-transcriptomic (MET) datasets. By directly fusing the three modalities together and assigning the label for each cell, UnitedNet identified the cell MET-types in an end-to-end manner with a high degree of congruence ($ARI = 0.82$) and a roughly diagonal correspondence ($ARI = 0.41$) between the major MET-types and subtle MET-types compared with the previously reported results (Fig. 3b-d). Furthermore, we benchmarked the performance of UnitedNet by comparing the results of MET-types clustering using including Schema¹⁷, MOFA³⁶, totalVI¹⁶, and WNN⁹ and Leiden clustering³⁷ using single modality data. The benchmarking results showed that UnitedNet performed similarly or better in terms of group identification accuracy compared to these methods (Supplementary Fig. 6). We also visualized learned modality-specific weights and found that the weights of gene expression were higher than other modalities in the Patch-seq dataset, which aligns with the previous conclusion that gene expression is a more informative modality in the Patch-seq dataset²⁹.”

“We first applied UnitedNet to a single-batch DBiT-seq embryo dataset, which simultaneously mapped the whole transcriptome and 22 proteins on an embryonic tissue³¹. Specifically, we generated the weighted average of RNA expression in the cell niche, which encodes spatial information, as the third modality for analysis. UnitedNet then combined gene expression, protein, and niche modalities for unsupervised joint identification of tissue regions and cross-modal prediction between gene expression and proteins. We benchmarked the accuracy of tissue region identification by considering the anatomic annotation of the tissue region from the original report as the ground truth³¹. UnitedNet achieved a higher unsupervised group identification accuracy compared with state-of-the-art methods, including Schema, MOFA, WNN, and single-modality Leiden clustering (Fig. 7b).

“Next, we applied UnitedNet to an annotated multi-batch spatial-omics dataset for simultaneous supervised joint group identification and cross-modal prediction. We used a human dorsolateral prefrontal cortex (DLPFC) dataset that spatially maps gene expression and H&E staining on 6-layered DLPFC brain slices from 12 batches³⁰. Similarly, we used the gene expression, H&E staining-based morphological features, and cell niche modalities as inputs to UnitedNet. UnitedNet can successfully annotate unseen DLPFC slices (Supplementary Fig. 16) and achieve higher accuracy than the other benchmarking method and an ablated version of UnitedNet that did not use the alternating training scheme or the cell niche modality (Fig. 7d and Supplementary Fig. 17). We also visualized learned modality-specific weights and found that the weights of gene expression were higher than other modalities in the spatial DLPFC dataset, which aligns with the previous conclusion that gene expression is a more informative modality in the DLPFC dataset.”

Comment #2: How do the neural interpretability methods used on UnitedNet compare to another multimodal integration technique that has built-in interpretability capabilities: Schema (<https://genomebiology.biomedcentral.com/articles/10.1186/s13059-021-02313-2>)?

Response to Comment #2: We thank the reviewer for this inspiring question. Schema is one of the *state-of-the-art* algorithms for multimodal data integration. The interpretable learning methods used in UnitedNet and Schema are different in their implementation methods and the learning outcomes, which are discussed as follows.

1. Implementation methods of UnitedNet and Schema are different. Schema used metric learning and UnitedNet used explainable learning. Metric learning in Schema rescales each feature in the primary modality so that the transformed data from the primary modality align optimally with data from the other modalities¹⁰. The feature-specific weights from the transformation indicate the influence of features in the primary modality in synthesizing cross-modal information. In contrast, explainable learning in UnitedNet evaluates the influence of each input feature to the model's outcome. Specifically, UnitedNet uses SHapley Additive exPlanations (SHAP)¹⁵ concept to assign importance values, known as Shapley value, to each input feature regarding to each model output (*e.g.*, a certain identified cell type or the cross-modality prediction of a certain feature). The Shapley value explains the contribution of a specific input feature by quantifying the difference between the output values with and without the effect of this feature.

2. Interpretable learning outcomes of UnitedNet and Schema are different. First, in terms of implementation, Schema requires the identification of a primary modality and outputs the relevance of each feature in that primary modality to all the other secondary modalities, while UnitedNet does not identify a primary modality and instead outputs the relevance of each feature to any model outcomes, such as features from other modalities or identified cell types and states.

Second, UnitedNet and Schema interpret multi-modality biology datasets from different viewpoints. UnitedNet is used to identify relationships between features of different modalities within a group (*e.g.*, cell types). For example, in the Patch-seq GABAergic neuron dataset, after training, the UnitedNet can be dissected using SHAP to quantify cell-type specific gene-to-function relationships. In contrast, Schema is used to identify the relevance between primary modality features and all secondary modalities as a whole. By designing specific secondary modalities, Schema can determine the importance of features in the primary modality to the secondary modalities. For example, when analyzing an scRNA-seq dataset that profiles neurons across the *Drosophila melanogaster* lifespan, Schema used gene expression data as the primary modality and neuron age as the secondary modalities. This allowed Schema to identify a set of age-related genes.

Revision to the manuscript: We included the following sentence in our introduction section: *“Schema represents state-of-the-art multi-modal integration methods that can identify the features in a user-defined primary modality that are important to other modalities¹⁷”*

In addition, we included a detailed experimental comparison in the unsupervised group identification task between Schema and UnitedNet in our **Response to Comment #1**.

Comment #3: Relatedly, how does UnitedNet compare against MOFA, totalVI, and Schema for unsupervised group identification? Some benchmarking has been done to compare UnitedNet against other methods for learning joint multimodal representations, but Schema has been shown to outperform several of these state-of-the-art methods in the unsupervised group identification task.

Response to Comment #3: We thank the reviewer for this comment. We agree with the reviewer that Schema should be included in the performance benchmarking. In the revised manuscript, we compared the performance of UnitedNet, MOFA, totalVI, and Schema on the unsupervised group identification task (**Response Figure 1-2**). Due to totalVI’s requirement for two-modality integer data as input, we were unable to compare its performance on the three-modality Patch-seq and four-modality DynGen datasets. The benchmarking results showed that UnitedNet had similar or better performance compared with these methods, with the exception that Schema outperforms on the Patch-seq subtype group identification task (**Response Figure 1-2 c**).

In addition, the benchmarking results also showed that Schema had a similar performance to UnitedNet and a better performance than MOFA, totalVI, and WNN. We thank the reviewer for this thoughtful question. The revision further strengthens our manuscript.

Revision to the manuscript: Please see our revision in **Response to Comment #1**.

Comment #4: The authors use SHAP to interpret trained UnitedNet models and to generate feature-cell type and feature-feature maps, but validation of these maps is currently limited to brief mentions of how certain aspects of these maps recapitulate a few prior biological findings. A more thorough and quantitative validation procedure would be helpful in building support for the effectiveness and utility of these maps (e.g., using the Shapley values as a predictor for marker genes and showing statistics for their predictability of these genes.). Similarly, how robust are the outputs of these neural interpretability methods to different model initializations or hyperparameter choices?

Response to Comment 4: We thank the reviewer for these very inspiring suggestions. We agree with the reviewer that it would be beneficial to validate the relevance relationships identified in our study through a quantitative procedure, rather than relying solely on previous biological studies. Based on the reviewer’s suggestion, we further validated the results by 1) quantifying the

effectiveness and utility of identified feature maps; and 2) using cross-validations to quantitatively confirm the robustness of the interpretable learning in UnitedNet to different initializations and hyperparameter choices. The results are as follows.

1. Effectiveness and utility of explainable learning in UnitedNet. Shapley values have been successfully utilized to interpret feature importance in various fields such as computer vision¹⁶, natural language processing¹⁷, and anesthesiology¹⁸. In this study, we sought to determine if Shapley values could be used in a similar manner to understand feature importance in multi-modality biology. To do this, we compared the Shapley values of marker features identified in previous studies with those of the same number of randomly chosen features from all features (**Response Figure 1-3a**). Given that marker features are expected to be more biologically relevant, we hypothesized that the Shapley values of these features would be higher than those of randomly chosen features. Our results supported this hypothesis as we found significantly higher Shapley values for marker genes compared with randomly chosen features in both the Patch-seq GABAergic neuron dataset¹⁹ (**Response Figure 1-3b**) and the multiome ATAC+Gene expression BMMCs dataset² (**Response Figure 1-3c**). These results suggest that Shapley values can be effectively used as an indicator of feature importance.

Following the reviewer's suggestion, we further used the Shapley values as a predictor for marker genes (**Response Figure 1-3d**). We used a linear classifier to identify the marker features directly from Shapley values. Our results showed that using the Shapley value as a predictor for marker genes prediction resulted in statistically higher predictability, with the prediction accuracy 0.72 ± 0.07 (mean \pm S.D.) for the Patch-seq GABAergic neuron dataset (**Response Figure 1-3e**) and 0.91 ± 0.03 (mean \pm S.D.) for the multiome ATAC+Gene expression BMMCs dataset (**Response Figure 1-3f**). These accuracies were significantly higher than the random guess accuracy of approximately 0.5. These results demonstrate the effectiveness of Shapley values in predicting marker features in multi-modality biology.

Response Figure 1-3 Effectiveness validation of explainable learning in UnitedNet. **a**, Schematics showing the comparison of Shapley values for previous studies and randomly selected features. The Shapley value was calculated for each feature by dissecting the trained model. **b-c**, Barplot showing the Shapley value of marker features from the previous studies^{2,7} and randomly selected features in the Patch-seq GABAergic neuron dataset (**b**) and multiome ATAC+gene expression BMMC dataset (**c**). The Shapley values of the same modality are converted into absolute values and normalized to the range of [0,1] for comparison. **d**, Schematics of using a linear classifier to predict whether a feature is a marker feature based on its Shapley value. Randomly selected features were used as control in the linear classifier. **e-f**, Barplot showing the Shapley value of marker features from the previous studies and randomly selected features in the Patch-seq GABAergic neuron dataset (**e**) and multiome ATAC+gene expression BMMC dataset (**f**). The Shapley values of the same modality are converted into absolute values and normalized to the range of [0,1] for comparison.

2. Robustness test of the SHAP-identified relevance relationships. As suggested by the reviewer, we conducted robustness tests on the outputs of the explainable learning of UnitedNet. Based on our model design, there are two sources of randomness in the interpretable learning outcomes: the inherent randomness of the SHAP method and the randomness introduced by

training the UnitedNet model on different data. We evaluated the robustness of our method by considering these two factors.

First, to validate the robustness of SHAP with different hyperparameters, we used the entire Patch-seq GABAergic neuron dataset to train a UnitedNet model and then applied SHAP to dissect the trained model. SHAP requires two types of data for the model dissection process¹⁵: “background data”, used to determine the reference output of the model, and “calculation data” used to generate the Shapley value by comparing it to the background data. The “background data” is the only adjustable hyperparameter in SHAP. In our robustness test, we applied 10 folds cross-validation to the Patch-seq GABAergic neuron dataset, using each fold in turn as the “calculation data” and the remaining 9 folds as the “background data” (so there are 10 cross-validation replications). We then counted the frequency of the top n features identified by Shapley values in each cross-validation replication (following previous studies, $n = 7$ for each cell type in the Patch-seq GABAergic neuron dataset). The results (**Response Figure 1-4**) showed that the top 7 frequently selected features were consistently identified across the 10 cross-validation replications, which indicates the robustness of SHAP with respect to its inherent randomness.

Then, to further validate the robustness of SHAP in UnitedNet, we considered both the inherent randomness in SHAP and the randomness introduced by training UnitedNet on different data. To do so, we used explainable learning to dissect UnitedNet models trained on different folds of cross-validation (**Response Figure 1-5**). For example, instead of using the entire Patch-seq GABAergic neuron dataset to train a single UnitedNet model, we divided the dataset into 10 folds and used each 9 folds to train a separate UnitedNet model and the remaining fold for testing. We then apply SHAP to dissect each model using the same methods described above. We counted the frequency of the top n features identified by Shapley values in each cross-validation replication. The results showed that explainable learning identified similar sets of features despite the inherent randomness of the SHAP method and the randomness introduced by training the UnitedNet model on different datasets. We also applied this validation to the multiome ATAC+Gene expression BMMCs dataset, using 13 folds of cross-validation across 7 cell types and two modalities. The results (**Response Figure 1-7**) showed that the top 20 features were consistently identified across the 13 folds of cross-validation, demonstrating the robustness of the method. The number of folds in the cross-validations (10 and 13) was chosen based on previous studies^{2,13,14}. Taken together, these results support the robustness of our method for dissecting the trained UnitedNet with SHAP.

Response Figure 1-4 Robustness validation of explainable learning by dissecting a fixed UnitedNet model trained with Patch-seq GABAergic neuron dataset. a-c, Barplot showing the frequency of SHAP-identified neuron-type specific genes (a), electrophysiological features (b), and morphological features (c). First, the UnitedNet was trained on the entire Patch-seq

GABAergic neuron dataset. Then, the Patch-seq GABAergic neuron dataset was evenly divided into 10 folds and each fold was used as input of SHAP to calculate the Shapley values. For each fold, the modality features with the top 7 largest Shapley values were selected. Finally, we counted the frequency of the selected features across the 10 folds and chose the top 7 most frequent features where the results are presented in the barplots.

Response Figure 1-5 Robustness validation of UnitedNet with explainable learning. Schematics showing the process of robustness test of explainable learning in UnitedNet using cross-validations. The UnitedNet models trained with different model initialization and random samples for the calculation of SHAP (by cross validation) were independently dissected by the explainable learning method. Each dissection can generate a set of cell-type-specific features. The robustness of the identified features was defined as their repetition rate in all cross-validations.

Response Figure 1-6 Robustness validation of explainable learning by dissecting the UnitedNet models across multiple folds of cross-validation in Patch-seq GABAergic neuron dataset. a-c, Barplot showing the frequency of SHAP-identified neuron-type specific genes (a), electrophysiological features (b), and morphological features (c) in 10 cross-validation

replications. The top 7 most frequent features were shown in each frequency barplot. The procedure is similar with the one for **Response Figure 1-4** except we in turn set aside a validation fold from the 10 folds cross-validation, retrain UnitedNet on the remaining nine folds, and evaluate SHAP values on the validation fold.

Response Figure 1-7 Robustness test of UnitedNet with explainable learning in multiome ATAC+gene expression BMMC dataset. Barplot showing the frequency of SHAP-identified cell-type specific genes (reds) and DNA accessibilities (greens) in 13 cross-validation replications. The top 20 most frequent features were shown in each frequency barplot. The procedure is similar to the one for **Response Figure 1-4** except that we in turn set aside a validation fold from the 13 folds cross-validation, retrain UnitedNet on the remaining 12 folds, and evaluate SHAP values on the validation fold.

Revisions to the manuscript: the results of **Response Figure 1-3**, **Response Figure 1-4**, **Response Figure 1-5**, **Response Figure 1-6**, and **Response Figure 1-7** have been updated in **Fig. 4**, **Fig. 6**, **Supplementary Fig. 8**, **Supplementary Fig. 9**, **Supplementary Fig. 10**, **Supplementary Fig. 12**, **Supplementary Fig. 11**, **Supplementary Fig. 14**, and **Supplementary Fig. 15**.

We have included the following edits to the revised manuscript (New texts are highlighted bold): *“We then dissected the trained UnitedNet using explainable learning, SHAP, to identify the feature relevance in the Patch-seq GABAergic neuron datasets. **Specifically, we used SHAP to assign the importance value, known as the Shapley value, to each input feature with respect to any given model output such as a certain identified cell group or the cross-modality prediction of a certain feature**³⁵. By definition, features with high Shapley values are influential. Therefore, we chose features based on their ranking of the Shapley values. To validate the robustness and effectiveness of this identification approach, we conducted the following experiments.*

To evaluate the robustness, we considered (i) the inherent randomness of the SHAP method and (ii) the randomness introduced by training the UnitedNet model on different data, which are two major sources of randomness in the interpretable learning outcomes. First, to validate the robustness of SHAP with different hyperparameters, we used the entire Patch-seq GABAergic neuron dataset to train a UnitedNet model. Then, we applied 10 folds cross-validation to the Patch-seq GABAergic neuron dataset to calculate Shapley values (so there are 10 cross-validation replications). We then counted the frequency of the top n features identified by Shapley values in each cross-validation replication (following previous study²⁹, n = 7 for each cell type in the Patch-seq GABAergic neuron dataset). The results (Supplementary Fig. 8) showed that the top 7 frequently selected features were consistently identified across the 10 cross-validation replications, which indicates the robustness of SHAP with respect to its inherent randomness.

Then, to further validate the robustness of SHAP in UnitedNet, we considered both the inherent randomness in SHAP and the randomness introduced by training UnitedNet on different data. Specifically, we used explainable learning to dissect UnitedNet models trained on different folds of cross-validation (Supplementary Fig. 9). Instead of using the entire Patch-seq GABAergic

neuron dataset to train a single UnitedNet model, we divided the dataset into 10 folds, used every 9 folds to train a separate UnitedNet model and the remaining fold for testing. We then applied SHAP to dissect each model using the same methods described above. We counted the frequency of the top 7 features identified by Shapley values in each cross-validation replication. The results showed that explainable learning identified similar sets of features despite the inherent randomness of the SHAP method and the randomness introduced by training the UnitedNet model on different datasets (Supplementary Fig. 10). Taken together, these results support the robustness of our method for dissecting the trained UnitedNet with SHAP.

Next, we quantitatively evaluated the effectiveness of the Shapley values and these SHAP-selected features (Supplementary Fig. 11). Given that neuron-subtype-specific features identified by previous studies using prior knowledge are expected to be more biologically relevant, we hypothesized that the Shapley values of these features would be higher than those of randomly chosen features (Supplementary Fig. 11a-b). Our results supported this hypothesis as we found significantly higher Shapley values for marker genes compared with randomly chosen features in the Patch-seq GABAergic neuron dataset (Supplementary Fig. 11c). Furthermore, we used the Shapley values as a predictor for marker genes (Supplementary Fig. 11d). We used a linear classifier to identify the marker features directly from Shapley values. Our results showed that using the Shapley value as a predictor for marker genes prediction resulted in statistically higher predictability, with the prediction accuracy 0.72 ± 0.07 for Patch-seq GABAergic neuron dataset (Supplementary Fig. 11e). This accuracy is significantly higher than the random guess accuracy of approximately 0.5. These results demonstrate the effectiveness of Shapley values in predicting group-specific features in multi-modality biology.

The above experiments showed the robustness and effectiveness of the Shapley value-based feature relevance identification. Based on the ranking and robustness of the Shapley value for each feature (Supplementary Figs. 8-10, see Methods), the final selected subset of neuron-subtype-specific genes, electrophysiological features, and morphological features, and quantified the neuron-subtype-specific, cross-modal feature-to-feature relevance are shown in Fig. 4 and Supplementary Fig. 12.”

“We further dissected the UnitedNet trained by the multiome ATAC+gene expression data for cell-type-specific, cross-modal feature-to-feature relevance (Fig. 6 and Supplementary Fig. 14). We examined the effectiveness and robustness of our explainable learning in the same way as discussed above. These results support the effectiveness and robustness of our method for dissecting the trained UnitedNet with SHAP (Supplementary Figs. 11 and 15).

We then explored the biological values of the explainable learning in UnitedNet for the multiome ATAC+gene expression dataset. Using CD8+ T cell type as an example (Fig. 6a), our results first identified a subset of genes (e.g., CD8A, A2M, LEF1, and NELL2) and chromatin accessibility sites (e.g., CD8A, DPP8, KDM2B, and KDM6B) that were differentially expressed

for the CD8+ T cell type (Fig. 6b and d-e). Within these genes and chromatin accessibility sites, the chromatin accessibility sites PROS1, KDM2B, and KDM6B exhibited stronger relevance to CD8+ T cell-specific genes, suggesting their critical roles in CD8+ T cell functions (Fig. 6c). The results align with the previous studies that the elevated expression level of PROS1 in the CD8+ T cell is a crucial regulatory signal that prevents overactive immune responses⁴¹. Meanwhile, the lack of KDM2B expression initiates T-cell leukemogenesis⁴². Notably, a recent study finds that KDM6B, a sibling gene of KDM2B, directly regulates the generation of CD8+ T cells by inducing chromatin accessibility in effector-associated genes⁴³. Our results further suggest that KDM2B may also play an important role in regulating the generation of CD8+ T cells.”

We have also included the following methods into the Methods Section:

“Robustness test of explainable learning using cross-validations

We tested the robustness of SHAP with respect to two sources of randomness in the interpretable learning outcomes: the inherent randomness of the SHAP method and the randomness introduced by training the UnitedNet model on different data.

For the inherent randomness, we first trained UnitedNet on the entire dataset to eliminate the randomness from model training. We then used k folds cross-validation to the dataset to calculate the Shapley values from the trained UnitedNet. The calculation of Shapley values (using the Deep SHAP algorithm) requires a “background dataset” to determine the reference output of the model and “calculation data” to generate the Shapley value by comparing it to the background data³⁵. In the k folds cross-validation, we took each fold in turn as the “calculation data” and the remaining k-1 folds as the “background data” (resulting in k cross-validation replications). We then counted the frequency of the top n features identified by Shapley values in each cross-validation replication. If the top n features are consistently identified across different cross-validation replications (with high frequency), we conclude that SHAP is robust with respect to its inherent randomness.

We also tested the robustness of SHAP with respect to both its inherent randomness and the training of UnitedNet. The procedure is similar with the above one except we retrained UnitedNet on the k-1 folds of the data. Then, in this robustness test, only the top n most frequent features across each cross-validation were selected as the final identified features.”

Comment #5: Related to the previous point, details regarding how Shapley value thresholds are chosen to ascertain sufficient relevance in feature-cell type or feature-feature maps appear to be missing in the text.

Response to Comment #5: We thank the reviewer for raising the question. In our previous manuscript, we identified the n most relevant features for each cell type based on the ranking of the absolute value of the Shapley value for each feature. In these cases, the value of n was determined based on the average number of marker features reported in the previous studies. For example, we selected the top 7 and top 20 features with the largest Shapley values for the Patch-seq and multiome ATAC+ gene expression BMMCs dataset, respectively, based on the average number of marker features identified in their previous studies^{2,7}.

In the revised manuscript, we have improved our method for identifying marker features based on the suggestion in **Comment #4** from the reviewer. Specifically, to test the robustness of the Shapley value thresholds, we used multiple folds cross-validation to select the top n **frequently chosen** features by the model (**Response Figure 1-8**). For each fold, SHAP was applied to dissect the UnitedNet models trained with the data from the remaining folds and selected a subset of candidate marker features with the n largest Shapley values. Then, we calculated the frequencies of the selected features across the multiple folds of cross-validation and chose the top n most frequently chosen ones. The value of n was still determined based on the average number of marker features reported in previous studies such as $n=7$ for the Patch-seq GABAergic neuron dataset and $n=20$ for the multiome ATAC+Gene expression BMMCs dataset.

We also evaluated the significance of the selected n from a statistical testing perspective. For example, we used the Patch-seq GABAergic neuron dataset to test the significance of the selected threshold $n = 7$ in the above setup (**Response Figure 1-6**). To that end, we compared the frequency of the selected 7th feature appearing in 10 folds with a reference distribution of frequencies obtained from some random datasets. Each random dataset is generated from the original dataset by randomly shuffling the subjects of each feature independently. The idea is that if the selected features are indeed influential, its 7th largest SHAP value will tend to be significantly larger than that calculated from a random dataset, which has no feature-to-feature and feature-to-cell group associations. If that is the case, we reject the null hypothesis that the n^{th} feature is selected only due to randomness.

The specific testing procedure is as follows. First, we randomly shuffled each column of the Patch-seq GABAergic neuron dataset so each row could contain the feature values from different cells and generated k random datasets. Second, we selected the first 7 features on each generated random dataset with the same method as the above and calculated the frequency of the 7th selected feature from 10 folds for each of the three modalities and considered cell groups. Next, we averaged the obtained frequencies from each random dataset and denoted them by c_1, \dots, c_k . We also calculated the averaged frequency of the 7th selected feature from 10 folds of the original dataset, denoted by c_0 . Then, we would reject the null hypothesis that not all of the seven selected features are significant if and only if c_0 is larger than the $(1 - \alpha)^{\text{th}}$ quantile of c_1, \dots, c_k , namely it falls into the -right tail of the reference distribution, for any prespecified significance level. The reference distribution is shown in **Response Figure 1-8c**, where the red area is the 0.05-right tail, and we used $k=100$. Since the average frequency calculated from the original dataset was 6.25, there was

significant evidence that the selected 7 features are influential under the significance level of 0.05. In practice, we may choose different thresholds for different cell groups and modalities. Then, the above testing procedure can be applied in each of these cases to help decide the threshold.

Response Figure 1-8. Feature selection for explainable learning in UnitedNet. **a**, Schematic showing the selection of features for explainable learning in UnitedNet based on the ranking of Shapley values. First, the UnitedNet model trained in each fold of cross-validation was dissected using the explainable learning method to determine the Shapley value of each modality feature for each model output. Then, the cell-type-specific features in each modality were ranked based on their Shapley values. The top n highest ranking features were selected as the initial selected features. **b**, Schematics showing the selection of features for explainable learning in UnitedNet based on frequency. First, the same process is repeated using different model initialization (different folds of training data for cross-validation) for n times. The top n most frequent features across all cross-validation were then selected as the final selected features. **c**, 7 cell-type-specific features selected for each modality in the Patch-seq dataset. The distribution curve was obtained from applying kernel density estimation to the frequencies from $k=100$ random datasets. The red area corresponds to the 0.05 rejection region for the null hypothesis that not all of the seven selected features are significant. The average frequency calculated from the original dataset is 6.25, which falls inside the rejection region.

Revision to the manuscript: We have updated the Methods Section as described in our **Response to Comment#4**.

Comment #6: While UnitedNet demonstrates superior performance in the group identification and cross-modal prediction tasks, the manuscript would benefit from a deeper exploration or discussion of the utility of these specific tasks in generating new biological insights. It is presently not made clear in the text how one can leverage UnitedNet’s performance gains on these tasks to contribute to discoveries outside of applying neural interpretability methods to dissect the trained model.

Response to Comment #6: We thank the reviewer for the suggestion. In this response, we have included further explorations and discussions on how UnitedNet can be used to uncover novel biological insights. First, we highlighted the biological values of UnitedNet to enable more flexible and comprehensive multi-modal data analyses. This capability includes taking an unlimited number of modalities for data analysis and considering spatial information as an additional modality for analysis. Using the Patch-seq dataset as an example, we showed that UnitedNet can simultaneously consider transcriptomic, electrophysiological, and morphological modalities in GABAergic neurons, revealing strong alignment of the cell clusters across these three latent spaces. Furthermore, using the spatial DLPFC dataset as an example, we demonstrated the utilization of spatial niche information as a modality for multimodal analysis by UnitedNet. Second, we showed that, by applying the trainable adaptive weighting scheme, UnitedNet can be used to evaluate the importance of each modality for group identification. Together, we discussed the potential of applying these technical advancements to identify novel cell types and continuously synthesize missing or unmeasurable modalities. Last but not the least, it is important to note that since the unique perspective of our work is to apply explainable learning to a network alternatively trained by the joint group identification and cross-modal prediction tasks, the performance improvement for these two tasks can collectively enhance our ability to dissect the model, discovering cell-type specific, cross-modal feature-to-feature relationships. Our detailed responses are as follows:

1. UnitedNet allows for more flexible and comprehensive multi-modal data inputs for analysis. With the advancement of single-cell multimodal and multi-omics technology, it is now possible to measure multiple modalities from the same sample. Therefore, tools that can analyze more than two modalities are needed. UnitedNet can handle an unlimited number of modalities and perform multiple tasks simultaneously such as multi-modal integration, joint group identification, cross-modal prediction, and cell-type specific cross-modal feature-to-feature relevance. This makes UnitedNet a valuable tool for conducting comprehensive multi-modal analysis, potentially leading to new biological discoveries.

1.1. UnitedNet revealed consistent latent space across three modalities of morphology, electrophysiology, and gene expression in the Patch-seq measurement. We used the UnitedNet to analyze the Patch-seq GABAergic neuron dataset, which measured morphology, electrophysiology, and gene expression in the same cortical neurons. Using a computational tool to align these multiple modalities in the same neurons offers the opportunity to enable accurate cross-modal data prediction and a unified identification of neuron types. Specifically, previous studies^{13,14} demonstrated that if an aligned latent representation can be extracted across multi-modality measurements that captures salient characteristics of neurons in the individual data modalities, two important biological outcomes could be i) one modality measurement (*e.g.*, electrophysiology) can be used to predict the others (*e.g.*, gene expression) and vice versa, and ii) the neuron identities can be jointly defined by multiple modalities for downstream neuron type identification. Moreover, the alignment level of the latent representations can help reveal the

homogeneity and heterogeneity across molecular, function, and morphological properties of individual neurons.

Previous method such as coupled autoencoder successfully aligned the transcriptomics and electrophysiology modalities in the latent space^{13,14}. However, since it used an alignment loss function designed between electrophysiology and transcriptomics modality, it does not apply to analyze this three-modality dataset, thus missing the information of the morphology modality for the joint analysis. In contrast, UnitedNet does not require explicit loss functions to perform alignment between modalities, thus allowing for an unlimited number of modalities as inputs. We applied UnitedNet on the three-modality Patch-seq GABAergic neuron dataset and examined the learnt latent space of three modalities. The result showed that transcriptomic and electrophysiological modalities are strongly aligned with each other (**Response Figure 1-9 a-b**), which aligns with previous studies. In addition, our results showed that the morphology modality is also aligned with the transcriptomics and electrophysiology modalities, with a relatively less level of alignment for the Pvalb neurons (**Response Figure 1-9 a-c**). This relatively less level of alignment further supported the previous finding that, even with similar gene expression profile, the Pvalb neurons are described as electrophysiological homogenous but morphologically diverse²⁰.

Aligning these three modalities further offers the opportunity to enable translation of individual measurements across three modalities and move towards a unified, multimodal view of cellular diversity. We performed cross-modal prediction between the modalities of transcriptomics, electrophysiology, and morphology using UnitedNet. Results in **Response Figure 1-9 d-e** showed that UnitedNet enabled the prediction of individual measurements across three modalities with high accuracy. Meanwhile, such latent space alignment enabled UnitedNet to identify the cell types of GABAergic neurons jointly from the aligned latent representations of all three modalities. We then integrated the modality-specific latent representations as the shared latent representations to predict morphological-electrophysiological-transcriptomic (MET) jointly defined cell types (**Response Figure 1-10a-b**) and found that UnitedNet-predicted cell types are consistent with cell types defined by previous study using prior knowledge⁷ (**Response Figure 1-10c**).

1.2. UnitedNet has demonstrated improved brain region identification through the incorporation of spatial niche information. Spatial omics is another important multimodal technology that enables the measurement of spatially resolved multi-omics information in intact tissues²¹⁻²³. However, spatial information is often not fully utilized when analyzing the spatial omics data for group identification tasks. UnitedNet is flexible enough to integrate a wide range of different modalities as inputs, including spatial information, as demonstrated in the analysis of DLPFC data using cell niche information (neighborhood gene expression information of each cell) as an additional modality (**Response Figure 1-11a**). The results showed that the performance of tissue region identification was significantly improved (**Response Figure 1-11b-c**). We believe that the application of UnitedNet to spatial omics data has the potential to identify or define region-specific cell types or architectures in these spatial-omics datasets.

2. UnitedNet allows for automated assignment of modality weights. UnitedNet can determine the importance of each modality in the group identification task through a trainable adaptive weighting scheme (**Response Figure 1-12a**). This scheme generates modality-specific codes, which are the low-dimensional representations of the input modality and combines them with trainable weights to create shared latent codes. The adaptive weights are optimized during the training of the entire network. One key feature of this scheme is its ability to assign lower weights to modalities with more noise, reducing their impact on the group identification results. The effectiveness of the adaptive weight scheme was validated using simulated data, and the learned adaptive weights were applied to data analysis in the manuscript.

2.1. Effective validation of the adaptive weighting scheme. We used simulated datasets with controllable noise levels to validate the effectiveness of the adaptive weighting in UnitedNet. Specifically, following a previous study²⁴, we simulated datasets with cell morphology and transcriptomics data, where the transcriptomics data contained different levels of noise. We applied UnitedNet to these simulated datasets and found that the modality weight of the transcriptomics modality increased as we decreased its noise level (**Response Figure 1-12b**). These results confirm that the adaptive weighting scheme used in UnitedNet is effective in identifying the importance of different modality to optimize group identification performance.

2.2. Adaptive weights in multi-modality biological datasets. We visualized the adaptive weight of each modality for the datasets used in our previous manuscript (**Response Figure 1-12 c-f**). We found that the weight of each modality was similar for datasets with equally informative modalities (*i.e.*, DynGen simulated dataset, **Response Figure 1-12ci-iv**, and multiome ATAC+gene expression BMMC dataset, **Response Figure 1-12d**). However, for the Patch-seq GABAergic neuron dataset (**Response Figure 1-12e**) and spatial DLPFC dataset (**Response Figure 1-12f**), the weight of gene expression was significantly higher than that of other modalities, consistent with our findings in **Response Figure 1-9 a-c** and previous studies, which have shown that gene expression is a more informative modality for group identification task in these datasets^{4,7}.

Response Figure 1-9 Tri-modality analyses of the Patch-seq GABAergic neuron enabled by UnitedNet. **a-c**, Latent space visualization of the modality-specific codes of the transcriptomics (**a**), electrophysiology (**b**), and morphology (**c**) in the Patch-seq GABAergic neuron dataset. All codes are color-coded by the joint morphology-electrophysiology-transcriptomics (MET)-types defined by the previous study. **d**, Heatmap comparing cross-modal predicted gene expression,

electrophysiological, and morphological features averaged over annotated cell types with the ground truth.

Response Figure 1-10 (from original Fig. 3) Unsupervised cell type identification of the Patch-seq GABAergic neuron enabled by UnitedNet. a-b, UMAP representations of the shared codes in the shared latent space that are color-coded by the joint morphology-electrophysiology-transcriptomics (MET)-types labeled by the reference (a) and identified by UnitedNet (b), where MET major cell type annotations are in i) and MET cell subtype annotations are in ii). **c**, Confusion matrices comparing i) joint major cell types and ii) cell subtypes between the reference labels and UnitedNet-identified labels.

Response Figure 1-11 (from original Extended Data Figs. 9-10) Utilization of spatial niche information in UnitedNet. **a**, Schematics of spatial-omics data analyses pipeline by UnitedNet. Spatial omics simultaneously measure spatially resolved multi-omics data in intact tissue networks. UnitedNet extracts the cell neighborhood information as an additional modality together with other modalities. **b**, The boxplot shows the ablation analysis of cell niche modality in the DLPFC dataset. UnitedNet trained with gene expression, morphology, and cell niche modality shows better performance in joint group identification tasks than the UnitedNet trained only with gene expression and morphology modality. **c**, The spots of each measured brain slice are colored by tissue region labels predicted by UnitedNet. For each sample, UnitedNet is trained on the

remaining 11 batches and is tested on this batch. Each UnitedNet successfully transfers the annotation of the reference brain slice region.

Response Figure 1-12 Adaptive weighting scheme in the UnitedNet. **a**, Schematics showing the adaptive weighting scheme of the joint group identification task in UntiedNet. Each modality was encoded to the latent space as modality-specific codes through corresponding encoders. Then, the modality-specific latent codes were adaptively weighted based on the noise level and fused as a shared latent code. The shared latent code is then used for unsupervised/supervised group identification through the grouping module. **b-f**, Boxplots showing the modality weight learned in UnitedNet for different modalities of the simulated MUSE dataset (**b**), simulated DynGen dataset (**c**), multiome ATAC+gene expression BMCC dataset (**d**), Patch-seq GABAergic neuron dataset (**e**), and spatial DLPCF dataset (**f**).

3. UnitedNet has the potential to provide new biological insights through its analysis of multimodal data.

3.1. Novel cell type identification. UnitedNet’s ability to identify groups of cells in an unsupervised manner can be used to discover novel cell types defined by multiple modalities measured from the same cells. Cell type identification is a significant direction of research in fields such as neuroscience and developmental biology²⁵. However, characterizing cell types can be challenging due to their diverse phenotypic properties at various levels. By adaptively and effectively integrating various modalities measured from the same cells, UnitedNet has the potential to improve the current cell typing system, discover new cell types that may not be identified using single- or two-modality methods, and uncover cell-type-specific phenotypes and functions.

3.2. Continuous prediction of unmeasurable modalities. The temporal evolution of certain modalities, such as gene expression and DNA accessibility, is important for understanding the cellular states that drive biological function and disease. However, it can be challenging to continuously measure these modalities due to the need for physical dissociation or tissue fixation. In contrast, some modalities, such as electrical activity, can be relatively easily measured continuously without disrupting the tissue. Recent technical advances have made it possible to simultaneously measure both types of characteristics at discrete time points (*e.g.*, gene expression and electrophysiology)^{26,27}. By using these joint measurements to train the model, UnitedNet could potentially be used to predict end-point modality (gene expression) from continuous measurements of other modality (electrical activity). The enhanced performance of UnitedNet may allow for the construction of a reliable predictive model, enabling the use of continuous prediction to generate new biological insights.

4. Using interpretable learning to discover cross-modal feature-to-feature relationships. We believe that interpretable learning is still important for understanding how UnitedNet’s improved performance on these tasks can contribute to discoveries. The alternative training of UnitedNet increases the reliability of interpretable learning methods i.e., SHAP, which are used to interpret the model’s predictions, and are only meaningful when the model produces accurate outputs (Response Figure 1-13). As the application of interpretable learning to dissect a multimodal trained network represents one of the major novelties of this work, we still briefly discuss potential examples of new biological insights that could be enabled by interpretable learning.

Application of interpretable learning to UnitedNet has the potential to be used to explore cell-type-specific gene regulation programs. Cell type-specific gene-to-function relationships are essential for both basic and translational biological and biomedical research, but it can be technically challenging to control and measure the gene expression and corresponding products (*e.g.*, proteins, functions) with cell-type specificity and high throughput (*e.g.*, precise control of thousands of

genes) in wet-lab experiments²⁸⁻³⁰. A computational approach to automatically infer such a relationship could be very useful²⁹. We envision that UnitedNet with interpretable learning could potentially be used as a computational tool to address this challenge. By identifying candidates for relevance relationships directly from multimodal data, we can further specifically test these candidates by experimental validation.

Response Figure 1-13 (from original Fig. 1) Interpretable multi-task learning with UnitedNet. **a-b**, Schematics showing the multi-task learning between joint group identification (**a**) and cross-modal prediction (**b**) by UnitedNet for single-cell multi-modality data. **c-d**, Schematics showing the application of explainable learning methods to dissect the trained UnitedNet for identifying group-to-feature relevance (**c**) and the cross-modal feature-to-feature relevance (**d**). First, an interpretable learning method dissects the encoders of a trained UnitedNet to identify the most relevant input features to each group (**c**). Then, within the grouped input features from (**c**), the explainable learning method further dissects the encoders and decoders of the trained UnitedNet to identify group-specific cross-modal feature-to-feature relevance (**d**).

Revisions to the manuscript: the results of **Response Figure 1-9**, **Response Figure 1-10**, **Response Figure 1-11**, and **Response Figure 1-12** have been updated in **Fig. 3**, **Supplementary Fig. 3**, **Supplementary Fig. 5**, **Supplementary Fig. 7**, **Supplementary Fig. 16**, and **Supplementary Fig. 17**.

We have included the following edits to the revised manuscript (New texts are highlighted bold): **“For the cross-modal prediction task, previous methods such as coupled autoencoder were limited for the prediction between two modalities since they used an alignment loss function**

designed between two modalities, which cannot be applied to analyze this three-modality dataset. In contrast, UnitedNet does not require explicit loss functions for modality alignment, thus allowing for the inclusion of an unlimited number of modalities as inputs. UnitedNet enabled the prediction of individual measurements across three modalities with high fidelity (Fig. 3e, Supplementary Fig. 6). We further examined the learned latent space of three modalities by the UnitedNet and found strong alignment between the transcriptomics and electrophysiology modalities (Supplementary Fig. 7a-b), which aligns with previous studies¹¹. In addition, we found that the morphology modality was also aligned with the transcriptomics and electrophysiology modalities, with a relatively less level of alignment for the Pvalb neurons (Supplementary Fig. 7c). This relatively less level of alignment further supported the previous finding that, although Pvalb neurons have similar gene expression profile, they exhibit both electrophysiological homogeneity and morphological diversity³⁹.”

“Spatial omics is another important multimodal technology that allows for the measurement of spatially resolved multi-omics information in intact tissues⁴⁴⁻⁴⁶. However, spatial information is often not fully utilized when analyzing spatial omics data for group identification tasks. UnitedNet is flexible enough to integrate a wide range of different modalities as inputs, including spatial information. As demonstrated below, UnitedNet can utilize cell niche information (neighborhood gene expression information of each cell) as an additional modality to identify biologically meaningful groups and enable cross-modal prediction (See Methods, Fig. 7a).”

“UnitedNet can successfully annotate unseen DLPFC slices (Supplementary Fig. 16) and achieve higher accuracy than the other benchmarking method and an ablated version of UnitedNet that did not use the alternating training scheme or the cell niche modality (Fig. 7d and Supplementary Fig. 17). We also visualized learned modality-specific weights and found that the weights of gene expression were higher than other modalities in the spatial DLPFC dataset, which aligns with the previous conclusion that gene expression is a more informative modality in the DLPFC dataset.”

“In summary, UnitedNet provides a new approach, which can extract spatial information as an input modality to enable both supervised and unsupervised group identification and cross-modal prediction for spatial-omics data.”

“We envision several directions in that UnitedNet could be used for data-driven scientific discoveries. In the group identification task, UnitedNet can adaptively and effectively integrate multiple modalities measured from the same cells, potentially improving current cell typing systems to discover new cell types that⁴⁷ may not be identified using single-modality methods and helping to infer cell-type-specific phenotypes and functions. For the cross-modal prediction task, UnitedNet may be able to predict end-point modality (e.g., gene expression) from continuous measurements of other modalities (e.g., electrical activity and bioimaging⁴⁸). The

enhanced performance of UnitedNet could enable the construction of a reliable predictive model, allowing for the use of continuous prediction to generate new biological insights. Last but not least, UnitedNet could be used to suggest the most likely relevance relationships from presented multi-modal data, providing useful guidance for downstream wet-lab validation and reducing time-consuming experiments.”

“In addition, we demonstrated the robustness of UnitedNet when handling datasets with modality-specific noise. In applications, noises arising from different sources, such as the sequencing dropout effect and feature measurement error in multi-modality biological datasets, typically affect the network's performance. To address this challenge, UnitedNet applied an adaptive weighting scheme to automatically assign lower weights to the modalities with more noise, reducing their impact on the group identification results (Supplementary Fig. 3a). To test the effectiveness of the adaptive weighting in UnitedNet, we used simulated datasets with controllable noise levels. Specifically, following a previous study²⁷, we simulated datasets with normal morphology modality data and noisy transcriptomics modality data at different controlled noise levels (Supplementary Fig. 3b) and applied UnitedNet to these datasets. The results showed that the modality weight of the transcriptomics modality increased as we decreased its noise level, enabling a similar performance compared with the state-of-the-art methods (Supplementary Fig. 3c-d). Meanwhile, in the simulated four-modality Dyngen dataset without noises, the modality-specific weights were similar across four modalities (Supplementary Fig. 3e).”

“Next, we performed simultaneous unsupervised joint group identification analysis and cross-modal prediction on the morphological-electrophysiological-transcriptomic (MET) datasets. By directly fusing the three modalities together and assigning the label for each cell, UnitedNet identified the cell MET-types in an end-to-end manner with a high degree of congruence ($ARI = 0.82$) and a roughly diagonal correspondence ($ARI = 0.41$) between the major MET-types and subtle MET-types compared with the previously reported results (Fig. 3b-d). Furthermore, we benchmarked the performance of UnitedNet by comparing the results of MET-types clustering using including Schema¹⁷, Multi-Omic Factor Analysis (MOFA)³⁶, totalVI¹⁶, and Weighted Nearest Neighbor (WNN)⁹ and Leiden clustering³⁷ using single modality data. The benchmarking results showed that UnitedNet performed similarly or better in terms of group identification accuracy compared to these methods (Supplementary Fig. 6). We also visualized learned modality-specific weights and found that the weights of gene expression were higher than other modalities in the Patch-seq dataset, which aligns with the previous conclusion that gene expression is a more informative modality in the Patch-seq dataset²⁹.”

“We applied UnitedNet to multiome ATAC+gene expression datasets measured from the bone marrow mononuclear cells (BMMCs) in 13 batches from different tissue sites and donors⁷. These datasets contain 21 previously identified and annotated cell types (Fig. 5b i-ii and Fig. 5c i-ii). We

trained UnitedNet using data from 12 batches of data with the corresponding cell type annotations serving as the labels from the training samples. We then tested the performance on the 13th batch. UnitedNet was able to successfully (i) integrate the 12-batch reference datasets and learned the 21 annotated cell types, (ii) simultaneously map the unlabeled test dataset (the 13th batch) into the same 21 separated clusters, and (iii) remove the batch effect (Fig. 5b iii-v and Fig. 5c iii). **We also visualized the learned modality-specific weights and found that the weights of DNA accessibility and gene expression were similar, indicating the similar importance of these two modalities with respect to cell type identification.**”

Minor Comment #1: There are a few points in the main text and figure captions where ATAC-seq is described as measuring both gene expression and chromatin accessibility in the same cell, but ATAC-seq refers only to the chromatin accessibility modality. This led to confusion about the experiments performed for the BBMCs dataset, which seem to profile both gene expression and chromatin accessibility modalities.

Response to Minor Comment #1: Thank you for bringing this to our attention. The BBMCs dataset used in our previous manuscript does indeed include both gene expression and chromatin accessibility modalities. To avoid any confusion, we have updated the name from “ATAC-seq” to “multiome ATAC+Gene expression”.

Revision to the manuscript: We have updated the name from “*ATAC-seq*” to “*multiome ATAC+Gene expression*” in the revised manuscript.

Minor Comment #2: The authors claim that UnitedNet’s unsupervised group classification predictions display a high degree of congruence with previously reported results. Heatmaps are provided to qualitatively show this congruence, but no statistics are shown in the text to provide quantitative support for this claim. Also, how do these unsupervised classification results compare with other state-of-the-art unsupervised classification methods?

Response to Minor Comment #2: We thank the reviewer for the suggestion. We fully agree that statistics should be provided to support our claim of high congruence in the Patch-seq dataset. To that end, we have calculated and highlighted the adjusted rand index (ARI) on top of the heatmaps (**Response Figure 1-14a**, ARI=0.82 for major cell types and ARI=0.41 for cell subtypes). Furthermore, as previously suggested by the reviewer and discussed in the **Response to Comment #1** and **Response to Comment #3**, we benchmarked the unsupervised classification performance of UnitedNet against other *state-of-the-art* methods. Specifically, in this Patch-seq dataset that the reviewer mentioned, UnitedNet still showed better performance in the major neuron type classification and similar performance in the subtype classification when compared with other methods (**Response Figure 1-14**).

Response Figure 1-14 Performance benchmarking of UnitedNet against the *state-of-the-art* methods in the Patch-seq dataset. a, Confusion matrices comparing joint major cell types and cell subtypes between the reference labels (adjusted rand index, ARI = 0.82) and UnitedNet-identified labels (ARI = 0.41). **b**, Barplot reporting the performance comparison between UnitedNet and other *state-of-the-art* methods for MET cell type identification using morphology, electrophysiology, and transcriptomic modalities in the Patch-seq GABAergic neuron dataset.

Revisions to the manuscript: the results of **Response Figure 1-14** have been updated in **Fig. 3**, and **Supplementary Fig. 6**.

We have included the following edits to the revised manuscript (New texts are highlighted bold): “Next, we performed simultaneous unsupervised joint group identification analysis and cross-modal prediction on the morphological-electrophysiological-transcriptomic (MET) datasets. By directly fusing the three modalities together and assigning the label for each cell, **UnitedNet identified the cell MET-types in an end-to-end manner with a high degree of congruence (ARI = 0.82) and a roughly diagonal correspondence (ARI = 0.41) between the major MET-types and subtle MET-types compared with the previously reported results (Fig. 3b-d). Furthermore, we benchmarked the performance of UnitedNet by comparing the results of MET-types clustering using including Schema¹⁷, Multi-Omic Factor Analysis (MOFA)³⁶, totalVI¹⁶, and Weighted Nearest Neighbor (WNN)⁹ and Leiden clustering³⁷ using single modality data. The benchmarking results showed that UnitedNet performed similarly or better in terms of group identification accuracy compared to these methods (Supplementary Fig. 6).**”

Minor Comment #3: Recent tools have been developed to infer feature-feature maps across single-cell modalities (e.g., GridNet, ICLR 2022, for peak-gene links) and should be referenced alongside the use of neural interpretability methods for generating such maps.

Response to Minor Comment #3: We thank the reviewer for introducing to us the new reference. We have cited the suggested reference in the revised manuscript.

Revision to the manuscript: *“More recently, GrID-NET has also been proposed to identify genomic loci that mediate the regulation of specific genes in multiome ATAC+gene expression datasets¹⁸”*

Response to Reviewer #2 Comments:

General Comment: This work by Tang etc. reported a multi-modality analysis pipeline that can integrate data from different modalities and further explore the relationship between them. The UnitedNet was validated and demonstrated impressive performance with quite a few datasets with different modalities, including Patch-seq, ATAC-seq, DBiT-seq, and spatial transcriptomics. This work is an important contribution and represents a significant advance to the multimodal data integration of high throughput genomics research, given the fact that more multiomics-techniques in single cell and spatial fields are developed.

Response to General comment: We thank the reviewer for these positive comments and thoughtful suggestions on our work. In the revised manuscript, we have published our GitHub repository, unified the metrics for performance evaluation throughout the manuscript, added visualizations of the learned modality weights, and provided more discussion about how alternating training may reduce the batch effects. We believe that these revisions suggested by the reviewer have greatly improved our paper.

Comment #1: The Github is not accessible for review at present. This has to be corrected in a revised submission and I expected to see the Github page when the revised version was received.

Response to Comment #1: We are sorry that the reviewer did not have access to the code. In the previous submission, as requested by the editor, we shared the code through the *Nature Communication* submission system and uploaded all the code as a zip file named UnitedNet_NatComm.zip. We are sorry if the reviewer did not receive the code. In the revised submission, we have made our GitHub repository publicly available at <https://github.com/LiuLab-Bioelectronics-Harvard/UnitedNet>.

Comment #2: Various metrics were used throughout the manuscript to compare the performance, can the authors justify the reason for switching the metric between different datasets? Will one metric contradict another metric regarding performance or accuracy?

Response to Comment #2: We thank the reviewer for pointing this out. We fully agree with the reviewer that the metrics should be used consistently and unbiasedly throughout the manuscript. As shown in **Response Table 1**, we have unified all the metrics as adjusted rand index (ARI) and coefficient of determination (R^2) to avoid confusion in the revised manuscript. However, we recognized that R^2 is a misleading metric for the binary-valued data³¹. Therefore, following a previous study³², we have used the area under the curve (AUC) as the metric for the binary-valued chromatin accessibility modality in the multiome ATAC+gene expression dataset.

As a result, with the updated metrics, UnitedNet still showed better performance compared with other *state-of-the-art* methods with the same metrics in the spatial-simulated (MUSE) dataset²⁴

(**Response Figure 2-1a**), multiome ATAC+gene expression dataset² (**Response Figure 2-1b**), spatial DBiT-seq dataset⁶ (**Response Figure 2-2**) and spatial DLPFC dataset (**Response Figure 2-3**).

Response Table 1 Metrics used in the revised manuscript. Table showing the metrics used to evaluate the performance of UnitedNet in each dataset in the revised manuscript. Unless requested by the characteristics of the dataset, the metrics have been unified as ARI and R2. The only exception is the use of AUC to evaluate the prediction of chromatin accessibility prediction, which is necessary due to its binary distribution.

Datasets	Joint group identification	Cross-modal prediction	Position in the previous manuscript
MUSE simulated dataset	ARI	N/A	Extended Data Fig. 3
Dyngen simulated datasets	ARI	R2	Fig 2, Extended Fig 2
Patch-seq gabaergic neuron dataset	ARI	R2	Fig 3, Extended Fig 4, 5, 6
Multiome ATAC+ gene expression	ARI	R2 and AUC	Fig 5
spatial DBiT-seq dataset	ARI	R2	Fig 7
spatial DLPFC dataset	ARI	R2	Fig.7, Extended Fig 10

Response Figure 2-1 Performance evaluation using consistent metrics in simulated datasets and multi-omics datasets. **a**, Barplot showing performance comparison between UnitedNet and *state-of-the-art* methods including single-modality Leiden clustering, Schema, WNN, MOFA, and totalVI on the DynGen simulated dataset with different dropout levels (0.001 to 1) in the joint group identification task. ARI was used to quantify the identification accuracy. **b**, Boxplot showing the performance comparison between the UnitedNet trained by alternating training, cross-modal

prediction task only, and joint group identification task only for the multiome ATAC+gene expression dataset. R^2 was used to quantify the cross-modal prediction accuracy.

Response Figure 2-2 Performance evaluation using consistent metrics in simulated datasets and multi-omics datasets. a, Barplot showing performance comparison between UnitedNet and *state-of-the-art* methods including single-modality Leiden clustering, Schema, WNN, and MOFA on the DBiT-seq embryo dataset in the joint group identification task. ARI was used to quantify the identification accuracy. **b,** Results of cross-modal prediction between representative genes and proteins from the DBiT-seq dataset. R^2 was used to quantify the cross-modal prediction performance.

Response Figure 2-3 Performance evaluation using consistent metrics in simulated datasets and multi-omics datasets. Prediction results of representative genes, and morphological features from the DLPFC dataset. R^2 was used to quantify the cross-modal prediction performance.

Revisions to the manuscript: the results of **Response Figure 2-1**, **Response Figure 2-1**, and **Response Figure 2-3** have been updated in **Fig. 3**, **Fig. 5**, **Fig. 7**, and **Supplementary Fig. 3**.

We have included the following edits to the revised manuscript (New texts are highlighted bold): “*We first applied UnitedNet to a single-batch DBiT-seq embryo dataset, which simultaneously mapped the whole transcriptome and 22 proteins on an embryonic tissue³¹. Specifically, we generated the weighted average of RNA expression in the cell niche, which encodes spatial information, as the third modality for analysis. UnitedNet then combined gene expression, protein, and niche modalities for unsupervised joint identification of tissue regions and cross-modal prediction between gene expression and proteins. **We benchmarked the accuracy of tissue region identification by considering the anatomic annotation of the tissue region from the original report as the ground truth³¹. UnitedNet achieved a higher unsupervised group identification accuracy compared with state-of-the-art methods, including Schema, MOFA, WNN, and single-modality Leiden clustering (Fig. 7b). In addition, UnitedNet enabled the spatially resolved cross-modal prediction between gene and protein expressions (gene expression-to-protein, $R^2 = 0.58 \pm 0.20$, protein-to-gene expression, $R^2 = 0.23 \pm 0.09$; Fig. 7c)**”*

“Furthermore, UnitedNet can enable the cross-modal prediction between the gene expression and the H&E morphological features (gene expression-to-H&E features, $R2 = 0.18 \pm 0.021$ and H&E features-to-gene expression, $R2 = 0.17 \pm 0.014$; Fig. 7e).”

Comment #3: Is there a way to show the weight of each modality for the clustering analysis? Quite like what is done in the WNN integration.

Response to Comment #3: We thank the reviewer for pointing this out. Similar to WNN integration¹¹, UnitedNet generates an adaptive weight of each modality for the clustering analysis (*i.e.*, the group identification task in our context). Specifically, UnitedNet used a trainable adaptive weighting scheme to evaluate the importance of each modality during group identification (**Response Figure 2-4a**). In this scheme, UnitedNet first generates modality-specific codes, which are the low-dimensional representations of the input modality, and then fuses these codes with trainable weights to create shared latent codes. The weights will be optimized during the training of the entire network. This scheme allows UnitedNet to adaptively assign lower weights to the modalities with more noise, reducing their impact to the group identification results.

We agree with the reviewer that the learned adaptive weights should be demonstrated in the manuscript. In the following discussion, we first validated the effectiveness of the adaptive weight scheme using simulated data. Then, we showed the learned adaptive weights for each dataset used in our manuscript.

1. Effective validation of the adaptive weighting scheme. To test the effectiveness of the adaptive weighting in UnitedNet, we used simulated datasets with controllable noise levels. Specifically, following a previous study²⁴, we simulated datasets with normal morphology modality data and noisy transcriptomics modality data at different controlled noise levels. We then applied the UnitedNet to the simulated datasets. The results showed that the modality weight of the transcriptomics modality increased as we decreased its noise level (**Response Figure 2-4b**). The results confirmed the effectiveness of the adaptive weighting scheme used in UnitedNet.

2. Adaptive weights in multi-modality biological datasets. We then visualized the adaptive weight of each modality for the datasets used in our previous manuscript (**Response Figure 2-4 c-f**). We found that the weight of each modality is comparable for datasets with equally informative modalities (*i.e.*, DynGen simulated dataset, **Response Figure 2-4ci-iv**, and multiome ATAC+gene expression BMMC dataset, **Response Figure 2-4d**). In contrast, for the Patch-seq GABAergic neuron dataset (**Response Figure 2-4e**) and spatial DLPFC dataset (**Response Figure 2-4f**), the weight of gene expression is significantly higher than that of other modalities, consistent with previous studies that showed gene expression is a more informative modality compared with the functional or morphological measurements in group identification task in these datasets^{4,7}.

Response Figure 2-4 Adaptive weighting scheme in the UnitedNet. **a**, Schematics showing the adaptive weighting scheme of the joint group identification task in UnitedNet. Each modality was encoded to the latent space as modality-specific codes through corresponding encoders. Then, the modality-specific latent codes were adaptively weighted based on the noise level and fused as a shared latent code. The shared latent code is then used for unsupervised/supervised group identification through the grouping module. **b-g**, Boxplots showing the modality weight learned in UnitedNet for different modalities of the simulated MUSE dataset (**b**), simulated DynGen dataset (**c**), multiome ATAC+gene expression BMMC dataset (**d**), Patch-seq GABAergic neuron dataset (**e**), and spatial DLPFC dataset (**f**).

Revisions to the manuscript: the results of **Response Figure 2-4** have been updated in **Supplementary Fig. 3**.

We have included the following edits to the revised manuscript (New texts are highlighted bold):
“*In addition, we demonstrated the robustness of UnitedNet when handling datasets with modality-specific noise. In applications, noises arising from different sources, such as the sequencing dropout effect and feature measurement error in multi-modality biological datasets, typically affect the network's performance. To address this challenge, UnitedNet applied an adaptive weighting scheme to automatically assign lower weights to the modalities with more noise, reducing their impact on the group identification results (Supplementary Fig. 3a). To test the effectiveness of the adaptive weighting in UnitedNet, we used simulated datasets with controllable noise levels. Specifically, following a previous study²⁷, we simulated datasets with normal morphology modality data and noisy transcriptomics modality data at different controlled noise levels (Supplementary Fig. 3b) and applied UnitedNet to these datasets. The results showed that the modality weight of the transcriptomics modality increased as we decreased its noise level, enabling a similar performance compared with the state-of-the-art methods (Supplementary Fig. 3c-d). Meanwhile, in the simulated four-modality Dyngen dataset without noises, the modality-specific weights were similar across four modalities (Supplementary Fig. 3e).*”

“*Next, we performed simultaneous unsupervised joint group identification analysis and cross-modal prediction on the morphological-electrophysiological-transcriptomic (MET) datasets. By directly fusing the three modalities together and assigning the label for each cell, UnitedNet identified the cell MET-types in an end-to-end manner with a high degree of congruence (ARI = 0.82) and a roughly diagonal correspondence (ARI = 0.41) between the major MET-types and subtle MET-types compared with the previously reported results (Fig. 3b-d). Furthermore, we benchmarked the performance of UnitedNet by comparing the results of MET-types clustering using including Schema¹⁷, Multi-Omic Factor Analysis (MOFA)³⁶, totalVI¹⁶, and Weighted Nearest Neighbor (WNN)⁹ and Leiden clustering³⁷ using single modality data. The benchmarking results showed that UnitedNet performed similarly or better in terms of group identification accuracy compared to these methods (Supplementary Fig. 6). We also visualized learned modality-specific weights and found that the weights of gene expression were higher than other modalities in the Patch-seq dataset, which aligns with the previous conclusion that gene expression is a more informative modality in the Patch-seq dataset²⁹.*”

““*UnitedNet can successfully annotate unseen DLPFC slices (Supplementary Fig. 16) and achieve higher accuracy than the other benchmarking method and an ablated version of UnitedNet that did not use the alternating training scheme or the cell niche modality (Fig. 7d and Supplementary Fig. 17). We also visualized learned modality-specific weights and found that the weights of gene expression were higher than other modalities in the spatial DLPFC dataset, which aligns*

with the previous conclusion that gene expression is a more informative modality in the DLPFC dataset.”

“We applied UnitedNet to multiome ATAC+gene expression datasets measured from the bone marrow mononuclear cells (BMMCs) in 13 batches from different tissue sites and donors⁷. These datasets contain 21 previously identified and annotated cell types (Fig. 5b i-ii and Fig. 5c i-ii). We trained UnitedNet using data from 12 batches of data with the corresponding cell type annotations serving as the labels from the training samples. We then tested the performance on the 13th batch. UnitedNet was able to successfully (i) integrate the 12-batch reference datasets and learned the 21 annotated cell types, (ii) simultaneously map the unlabeled test dataset (the 13th batch) into the same 21 separated clusters, and (iii) remove the batch effect (Fig. 5b iii-v and Fig. 5c iii). We also visualized the learned modality-specific weights and found that the weights of DNA accessibility and gene expression were similar, indicating the similar importance of these two modalities with respect to cell type identification.”

Comment #4: The batch effects were indeed corrected with UnitedNet. But how this alternating training feature can reduce the batch effects is not clearly explained or analyzed.

Response to Comment #4: We appreciate the reviewer for raising the point and agree that more discussion and analyses on the batch effect correction in UnitedNet are necessary. In response, we conducted additional ablation studies to provide insights into the key components of UnitedNet for batch effect correction. Our ablation studies showed that while our joint group identification task helps remove the batch effect through its classification loss function, it also leads to overfitting of group identification. In contrast, the cross-modal prediction task improves group identification, but can also contribute to a stronger batch effect. By combining both tasks in an alternating training approach, we were able to leverage their strengths to reduce the batch effect while maintaining group identification performance. This result is consistent with previous studies using the prediction loss³³⁻³⁵. Our detailed ablation studies and results are as follows.

To investigate the mechanism for batch effect correction in UnitedNet, we conducted an ablation study with the following three approaches: 1) training UnitedNet only with the cross-modal prediction task, 2) training UnitedNet only with the joint group identification task, and 3) alternately training UnitedNet with both tasks. We applied this ablation study to both the multiome ATAC+gene expression BMMCs dataset² (**Response Figure 2-5**) and the spatial DLPFC dataset⁴ (**Response Figure 2-6**).

We visualized the latent space of UnitedNet trained with only the group identification task or only the cross-modal prediction task. We found that classification loss in the group identification task effectively removes the batch effect in the latent space in both multiome ATAC+gene expression BMMCs dataset (**Response Figure 2-5a**) and DLPFC dataset (**Response Figure 2-6a**). However, the results also showed lower separability with several groups mixed together (**Response Figures**

2-3d and **2-4d**), leading to poor group identification performance (**Response Figure 2-5e** and **2-4e**). This suggests that UnitedNet trained with only the group identification task may potentially overfit the training data. In contrast, we found that the cross-modal prediction task generates a mixed latent space with strong batch effects (**Response Figures 2-3b** and **2-4b**). Finally, the latent space of UnitedNet trained with alternating tasks (**Response Figures 2-3c** and **2-4c**) showed that the alternating training approach maintains both the ability to remove the batch effect and good separability in the latent space (**Response Figures 2-3d** and **2-4d**). As a result, alternating training resulted in the highest performance in the group identification task (**Response Figures 2-3e** and **2-4e**), suggesting that it also reduced the overfitting.

Response Figure 2-5 Batch effect removal in multiome ATAC+gene expression BMBCs dataset enabled by UnitedNet. a-c, The UMAP latent space visualizations showing shared latent codes of UnitedNet trained with group identification task only (a), cross-modal prediction task only (b), and alternating training (c) of the 13-batch multiome ATAC+gene expression BMBCs dataset. In the training manifold, the latent codes are colored by different batches. In the testing manifold, the latent codes are colored by cell-type annotations. **d-e,** Boxplot showing the comparison of testing manifold separability in terms of Silhouette score and testing data group identification accuracy in terms of ARI between the UnitedNet trained by joint group identification

task only (d), cross-modal prediction task only (e), and alternating training (f) for the multiome ATAC+gene expression dataset.

Response Figure 2-6 Batch effect removal in spatial DLPFC dataset enabled by UnitedNet. a-c, The UMAP latent space visualizations showing shared latent codes of UnitedNet trained with group identification task only (a), cross-modal prediction task only (b), and alternating training (c) of the 11-batch spatial DLPFC dataset. In the training manifold, the latent codes are colored by different batches. In the testing manifold, the latent codes are colored by cell-type annotations. d-f, Boxplot showing the comparison of testing manifold separability in terms of Silhouette score

and testing data group identification accuracy in terms of ARI between the UnitedNet trained by joint group identification task only (**d**), cross-modal prediction task only (**e**), and alternating training (**f**) for the spatial DLPFC dataset.

Revisions to the manuscript: the results of **Response Figure 2-5** and **Response Figure 2-6** have been updated in **Supplementary Fig. 13** and **Supplementary Fig. 18**.

We have included the following edits to the revised manuscript (New texts are highlighted bold):
“*We applied UnitedNet to analyze complex and large-scale datasets from multiple batches of samples. These large-scale multi-omics datasets are typically generated to annotate cell molecular types from different batches of samples. For example, the single-cell transposase-accessible chromatin sequencing (ATAC) technique that combines gene expression and genome-wide DNA accessibility (namely, multiome ATAC+gene expression data) has been used to profile diverse types of immune cells⁵. **One challenge in analyzing these datasets is using previously annotated multi-modality datasets as references to analyze new measurements from a different batch of the same biological system. Another challenge is the batch effect, which refers to differences between cells caused by inter-sample variations. This batch effect makes it difficult to use labeled multiome ATAC+gene expression datasets as reference atlases to annotate other new datasets. We found that UnitedNet can address these challenges in two ways. First, it can perform a supervised group identification task (termed annotation transfer) that automatically identifies cell types in new, unlabeled test samples based on previously labeled training samples, while simultaneously enabling cross-modal prediction. In addition, we found that the combination of supervised group identification with alternating training can effectively reduce batch effects among different biological samples (Fig. 5a).***”

“*To understand how UnitedNet removes batch effects, we conducted additional ablation studies. These studies showed that while the joint group identification task helped remove the batch effect through its classification loss function, it also led to overfitting of group identification (Supplementary Fig. 13 a, d). In contrast, the cross-modal prediction task improved group identification, but it could also contribute to a stronger batch effect (Supplementary Fig. 13 b, e). By combining both tasks in an alternating training approach, UnitedNet was able to leverage their strengths to reduce the batch effect while maintaining group identification performance (Supplementary Fig. 13c, f).*”

“*Additionally, we explored whether UnitedNet can remove the batch effect in the spatial DLPFC dataset, same as the analyses for the multiome ATAC+gene expression BMMC dataset. The results showed that UnitedNet maintained both good separability in the latent space and the ability to remove the batch effect, enabling the highest performance in the group identification task compared to other ablation studies (Supplementary Fig. 18). Furthermore, UnitedNet enabled cross-modal prediction between the gene expression and the H&E morphological features (gene expression-to-H&E features, $R^2 = 0.18 \pm 0.021$ and H&E features-to-gene expression, $R^2 = 0.17 \pm 0.014$; Fig. 7e).*”

References

- 1 Xu, C. *et al.* Probabilistic harmonization and annotation of single-cell transcriptomics data with deep generative models. *Mol. Syst. Biol.* **17**, e9620 (2021).
- 2 Luecken, M. D. *et al.* in *Thirty-fifth Conference on Neural Information Processing Systems Datasets and Benchmarks Track (Round 2)*.
- 3 Svensson, V., Teichmann, S. A. & Stegle, O. SpatialDE: identification of spatially variable genes. *Nat. Methods* **15**, 343-346 (2018).
- 4 Maynard, K. R. *et al.* Transcriptome-scale spatial gene expression in the human dorsolateral prefrontal cortex. *Nat. Neurosci.* **24**, 425-436 (2021).
- 5 Cannoodt, R., Saelens, W., Deconinck, L. & Saeys, Y. Spearheading future omics analyses using dyngen, a multi-modal simulator of single cells. *Nat. Commun.* **12**, 1-9 (2021).
- 6 Liu, Y. *et al.* High-Spatial-Resolution Multi-Omics Sequencing via Deterministic Barcoding in Tissue. *Cell* **183**, 1665-1681.e1618 (2020).
- 7 Gouwens, N. W. *et al.* Integrated morphoelectric and transcriptomic classification of cortical GABAergic cells. *Cell* **183**, 935-953. e919 (2020).
- 8 Argelaguet, R. *et al.* MOFA+: a statistical framework for comprehensive integration of multi-modal single-cell data. *Genome Biol.* **21**, 111 (2020).
- 9 Gayoso, A. *et al.* Joint probabilistic modeling of single-cell multi-omic data with totalVI. *Nat. Methods* **18**, 272-282 (2021).
- 10 Singh, R., Hie, B. L., Narayan, A. & Berger, B. Schema: metric learning enables interpretable synthesis of heterogeneous single-cell modalities. *Genome Biol.* **22**, 1-24 (2021).
- 11 Stuart, T. *et al.* Comprehensive integration of single-cell data. *Cell* **177**, 1888-1902. e1821 (2019).
- 12 Traag, V. A., Waltman, L. & van Eck, N. J. From Louvain to Leiden: guaranteeing well-connected communities. *Sci. Rep.* **9**, 5233 (2019).
- 13 Gala, R. *et al.* A coupled autoencoder approach for multi-modal analysis of cell types. *NeurIPS* **32** (2019).
- 14 Gala, R. *et al.* Consistent cross-modal identification of cortical neurons with coupled autoencoders. *Nat. Comput. Sci.* **1**, 120-127 (2021).
- 15 Lundberg, S. M. & Lee, S.-I. A unified approach to interpreting model predictions. *NeurIPS* **30** (2017).
- 16 Tjoa, E. & Guan, C. A survey on explainable artificial intelligence (xai): Toward medical xai. *IEEE Trans. Neural Netw. Learn. Syst.* **32**, 4793-4813 (2020).
- 17 Danilevsky, M. *et al.* A survey of the state of explainable AI for natural language processing. Preprint at <https://doi.org/10.48550/arXiv.2010.0071> (2020).
- 18 Lundberg, S. M. *et al.* Explainable machine-learning predictions for the prevention of hypoxaemia during surgery. *Nat. Biomed. Eng.* **2**, 749-760 (2018).
- 19 Gouwens, N. W. *et al.* Integrated Morphoelectric and Transcriptomic Classification of Cortical GABAergic Cells. *Cell* **183**, 935-953.e919 (2020).
- 20 Jiang, X. *et al.* Principles of connectivity among morphologically defined cell types in adult neocortex. *Science* **350**, aac9462 (2015).
- 21 Method of the Year 2019: Single-cell multimodal omics. *Nat. Methods* **17**, 1-1 (2020).
- 22 Wang, X. *et al.* Three-dimensional intact-tissue sequencing of single-cell transcriptional states. *Science* **361**, eaat5691 (2018).

- 23 He, Y. *et al.* ClusterMap for multi-scale clustering analysis of spatial gene expression. *Nat. Commun.* **12**, 1-13 (2021).
- 24 Bao, F. *et al.* Integrative spatial analysis of cell morphologies and transcriptional states with MUSE. *Nat. Biotechnol.* **1**, 10 (2022).
- 25 Zeng, H. What is a cell type and how to define it? *Cell* **185**, 2739-2755 (2022).
- 26 Li, Q. *et al.* Multimodal Charting of Molecular and Functional Cell States via in situ Electro-Seq.
- 27 Xu, S. *et al.* Behavioral state coding by molecularly defined paraventricular hypothalamic cell type ensembles. *Science* **370**, eabb2494 (2020).
- 28 Sylwestrak, E. L. *et al.* Cell-type-specific population dynamics of diverse reward computations. *Cell* **185**, 3568-3587. e3527 (2022).
- 29 Singh, R., Wu, A. P. & Berger, B. Granger causal inference on dags identifies genomic loci regulating transcription. Preprint at <https://doi.org/10.48550/arXiv.2210.10168> (2022).
- 30 Michaels, Y. S. *et al.* Precise tuning of gene expression levels in mammalian cells. *Nat. Commun.* **10**, 1-12 (2019).
- 31 Cox, D. R. & Wermuth, N. A comment on the coefficient of determination for binary responses. *Am Stat.* **46**, 1-4 (1992).
- 32 Wu, K. E., Yost, K. E., Chang, H. Y. & Zou, J. BABEL enables cross-modality translation between multiomic profiles at single-cell resolution. *Proc. Natl. Acad. Sci. U.S.A.* **118** (2021).
- 33 Robert, T., Thome, N. & Cord, M. in *Proceedings of the European Conference on Computer Vision (ECCV)*. 153-169.
- 34 Li, C., Liu, C., Duan, L., Gao, P. & Zheng, K. Reconstruction regularized deep metric learning for multi-label image classification. *IEEE Trans. Neural Netw. Learn. Syst.* **31**, 2294-2303 (2019).
- 35 Zhu, Q. & Zhang, R. A classification supervised auto-encoder based on predefined evenly-distributed class centroids. Preprint at <https://doi.org/10.48550/arXiv.1902.00220> (2019).

REVIEWERS' COMMENTS

Reviewer #2 (Remarks to the Author):

I am satisfied with the revised version

Point-by-Point Response to Reviewers' Comments, Manuscript #NCOMMS-22-34542A:

We are very grateful to the editor and reviewers for taking the time to review our work and for the positive feedback provided. We appreciate the opportunity to further address the reviewers' comments and to further improve and refine our research.

Response to Reviewer #2 Comment:

Comment: I am satisfied with the revised version.

Response: We appreciate the positive feedback from the reviewer and the valuable insights and suggestions offered. Thank you for taking the time to review our work!